# Is This the Subspace You Are Looking for? An Interpretability Illusion for Subspace Activation Patching

**Aleksandar Makelov & Georg Lange** [*]
SERI MATS
Constellation Offices
Berkeley, California, USA
`aleksandar.makelov@gmail.com`
`georglange4@gmail.com`

**Atticus Geiger**
Linguistics Department
Stanford University
Stanford, CA 94305, USA
`atticusg@stanford.edu`

**Neel Nanda**
`neelnanda27@gmail.com`

## Abstract

Mechanistic interpretability aims to attribute high-level model behaviors to specific, interpretable learned features. It is hypothesized that these features manifest as directions or low-dimensional subspaces within activation space. Accordingly, recent studies have explored the identification and manipulation of such subspaces to reverse-engineer computations, employing methods such as activation patching. In this work, we demonstrate that naïve approaches to subspace interventions can give rise to interpretability illusions.

Specifically, even if patching along a subspace has the intended end-to-end causal effect on model behavior, this effect may be achieved by activating *a dormant parallel pathway* using a component that is *causally disconnected* from the model output. We demonstrate this in a mathematical example, realize the example empirically in two different settings (the Indirect Object Identification (IOI) task and factual recall), and argue that activating dormant pathways ought to be prevalent in practice. In the context of factual recall, we further show that the illusion is related to rank-1 fact editing, providing a mechanistic explanation for previous work observing an inconsistency between fact editing performance and fact localisation.

However, this does not imply that activation patching of subspaces is intrinsically unfit for interpretability. To contextualize our findings, we also show what a success case looks like in a task (IOI) where prior manual circuit analysis informs an understanding of the location of a feature. We explore the additional evidence needed to argue that a patched subspace is faithful.

## 1 Introduction

Recently, large language models (LLMs) have demonstrated impressive (Vaswani et al., 2017; Devlin et al., 2019; OpenAI, 2023; Radford et al., 2019; Brown et al., 2020), and often surprising (Wei et al., 2022), capability gains. However, they are still widely considered 'black boxes': their successes – and failures – remain largely a mystery. It is thus an increasingly pressing scientific and practical question to understand *what* LLMs learn and *how* they make predictions.

This question is in the realm of machine learning interpretability, an important but notoriously slippery concept. Desiderata for interpretability are often not stated precisely, and it is easy to develop an *illusory* perception of understanding (Lipton, 2016; Adebayo et al., 2018; Bolukbasi et al., 2021). Mechanistic interpretability (MI) is a subfield of interpretability that seeks to avoid these pitfalls by developing a rigorous low-level understanding of the learned algorithms behind a model's computations. Specifically, MI frames these computations as collections of narrow, task-specific algorithms

---

[*]Equal contribution; full version at `https://arxiv.org/abs/2311.17030`

– *circuits* (Olah et al., 2020; Geiger et al., 2021; Wang et al., 2023) – whose operations are grounded in concrete, atomic building blocks akin to variables in a computer program (Olah, 2022) or causal model (Vig et al., 2020; Geiger et al., 2023a). MI has found applications in several downstream tasks: removing toxic behaviors from a model while otherwise preserving performance by minimally editing model weights (Li et al., 2023b), changing factual knowledge encoded by models in specific components to e.g. enable more efficient fine-tuning in a changing world (Meng et al., 2022a), improving the truthfulness of LLMs at inference time via efficient, localized inference-time interventions in specific subspaces (Li et al., 2023a) and studying the mechanics of gender bias in language models (Vig et al., 2020).

A central question in MI is: what *is* the proper definition of 'building blocks'(Olah, 2022)? Many initial mechanistic analyses have focused on mapping circuits to collections of *model components* (Wang et al., 2023; Heimersheim & Janiak). A workhorse of these analyses is *activation patching*[1] (Vig et al., 2020; Geiger et al., 2020; Meng et al., 2022a; Wang et al., 2023), which swaps component activations between examples and looks for task-relevant changes in model outputs. However, a plethora of empirical evidence suggests that the features LLMs represent and use lie in *linear subspaces* of component activations (Nanda, 2023a; Li et al., 2021; Abdou et al., 2021; Grand et al., 2018). Furthermore, phenomena like superposition and polysemanticity (Elhage et al., 2022) suggest that these subspaces are not easily enumerable, like individual neurons, but can rather be identified with a continuous space of arbitrary rotations – so searching for them can be non-trivial. This raises the question: can we carry over the success of activation patching from component-level analysis to finding the precise subspaces corresponding to features?

Indeed, recent works such as Geiger et al. (2023b); Wu et al. (2023) have begun identifying interpretable subspaces using gradient descent, with training objectives created using subspace activation patching. While this kind of end-to-end optimization has promise, we show that it is prone to a kind of *interpretability illusion*. Specifically, instead of robustly localizing a variable that is used by the model in a wide range of contexts, setting the value of a subspace can fabricate such a variable by activating a dormant pathway in the model via exploiting a causally disconnected feature (Figure 1). Our results suggest this effect is strongest when overfitting to a small dataset. Our contributions can be summarized as follows:

- In Section 3, we construct a distilled mathematical example of the illusion;

- In Section 4, we find a realization of this phenomenon 'in the wild', in the context of the indirect object identification task (Wang et al., 2023), where a 1-dimensional subspace of MLP activations found using DAS (Geiger et al., 2023b) can seem to encode position information about names in the sentence;

- To contextualize our results, in Section 5 we also show how DAS can be used to find subspaces that faithfully represent a feature in a model's computation. Specifically, we find a 1-dimensional subspace encoding the same position information in the IOI task, and validate its role in model computations via experiments beyond end-to-end causal effect. We argue that activation patching on subspaces of the residual stream is safer and less prone to illusions.

- Going beyond the IOI task, in Section 6 we also exhibit this phenomenon in the setting of *fact editing* (Meng et al., 2022a). We show that 1-dimensional activation patches imply equivalent rank-1 model edits (Meng et al., 2022a). In particular, this shows that rank-1 model edits can also be achieved by activating a dormant pathway in the model, without relying on the presence of a fact in the weight being edited. This suggests a mechanistic explanation for the observation of (Hase et al., 2023) that rank-1 model editing works regardless of whether the fact is present in the weights being edited.

- in Section 7, we end with arguments and evidence for why this interpretability illusion ought to be prevalent in real-world language models.

## 2   RELATED WORK

---

[1]also known as 'interchange intervention' (Geiger et al., 2020) and related to, but distinct from 'resample ablation' (Chan et al.) or 'causal tracing' (Meng et al., 2022a)

**Discovering and causally intervening on representations with activation patching**. *Activation patching* (Vig et al., 2020; Geiger et al., 2020) is a widely used causal intervention, whereby the model is run on an input A, but chosen activations are 'patched in' from input B. Motivated by causal mediation analysis (Pearl, 2001), activation patching has been used to localize model components causally involved in various behaviors, such as gender bias (Vig et al.), factual recall (Meng et al., 2022a), multiple choice questions (Lieberum et al., 2023), arithmetic (Stolfo et al., 2023) natural language reasoning (Geiger et al., 2021; Wang et al., 2023; Geiger et al., 2023b; Wu et al., 2023), code (Heimersheim & Janiak), and (in certain regimes) topic/sentiment/style of free-form natural language (Turner et al., 2023).

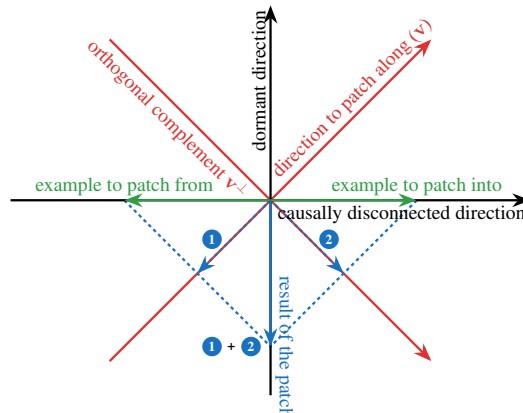

Figure 1: The key mathematical phenomenon behind the activation patching illusion (see Appendix Figure 26 for a step-by-step explanation). By setting the projection of an example's activation (green, right) along a vector (red, top-right) to equal another's (green, left) projection, we obtain a vector orthogonal to both activations (blue, down). This can give counterintuitive results when the original and new directions have fundamentally different roles in a model's computation.

Activation patching is an area of active research, and many recent works have extended the method, with patching paths between components (Goldowsky-Dill et al., 2023), automating the finding of sparse subgraphs (Conmy et al., 2023), fast approximations (Nanda, 2023b), and automating the verification of hypotheses (Chan et al.). In particular, a wide range of interpretability work (Mikolov et al., 2013; Conneau et al., 2018; Tenney et al., 2019; Burns et al., 2022; Nanda et al., 2023) suggests the *linear representation hypothesis*: models encode features as linear subspaces of component activations that can be arbitrarily rotated with respect to the standard basis (due to phenomena like superposition, polysemanticity (Arora et al., 2018; Elhage et al., 2022) and lack of privileged bases (Smolensky, 1986; Elhage et al., 2021)). Motivated by this, recent work such as Geiger et al. (2023b); Wu et al. (2023); Lieberum et al. (2023) has generalized activation patching to operate only on linear subspaces of features rather than patching entire components (heads, layers and neurons). Our work contributes to this research direction by demonstrating both (i) a common illusion to avoid when looking for such subspaces and (ii) a detailed case study of successfully localizing a binary feature to a 1-dimensional subspace.

**Factual recall**. A well-studied domain for discovering and intervening on learned representations is the localization and editing of factual knowledge in language models (Geva et al., 2023; Meng et al., 2022b; Wallat et al., 2020; Dai et al., 2022; Hernandez et al., 2023). A work of particular note is Meng et al. (2022a), which localizes and edits factual information with a rank-1 intervention on model weights. However, recent work has shown that rank-1 editing can work even on weights where the fact supposedly is not encoded (Hase et al., 2023), and that editing a single fact often fails to have its expected common-sense effect on logically related downstream facts (Cohen et al., 2023; Zhong et al., 2023). We contribute to this line of work by showing a formal and empirical connection between activation patching along 1-dimensional subspaces and rank-1 model editing. In particular, rank-1 model edits can work by creating a dormant pathway of an MLP layer, regardless of whether the fact is stored there. This provides a mechanistic explanation for the discrepancy observed in Hase et al. (2023).

**Interpretability illusions**. Despite the promise of interpretability, a common theme in the field is identifying ways that techniques may lead to misleading conclusions about of model behavior (Lipton, 2016). In computer vision, Adebayo et al. (2018) show that a popular class of pixel attribution methods is not sensitive to whether or not the model used to produce is has actually been trained or not. In Geirhos et al. (2023), the authors show how a circuit can be hardcoded into a learned model so that it fools interpretability methods; this bears some similarity to our illusion. In natural language processing, Bolukbasi et al. (2021) show that interpreting single neurons with

maximum activating dataset examples may lead to conflicting results across datasets due to subtle polysemanticity (Elhage et al., 2022).

## 3 A CONCEPTUAL VIEW OF THE ILLUSION

**Activation patching**. *Activation patching* (Vig et al., 2020; Geiger et al., 2020; Wang et al., 2023; Chan et al.) is an interpretability technique that intervenes upon model components, forcing them to take on values they would have taken if a different input were provided. For instance, consider a model that has knowledge of the locations of famous landmarks, and completes e.g. the sentence $A$ = 'The Eiffel Tower is in' with 'Paris'. How can we find which component of the model is responsible for knowing that 'Paris' is the right completion? Activation patching approaches this question by (i) running the model on $A$, (ii) storing the activation of a chosen component $c$, and (iii) running the model on e.g. $B$ = 'The Colosseum is in', *but* with the activation of $c$ taken from $A$. If we find that the model outputs 'Paris' instead of 'Rome' in step (iii), this suggests that component $c$ is important for the task of recalling the location of a landmark.

**Subspace Activation Patching**. The linear representation hypothesis proposes that *linear subspaces* of vectors will be the most interpretable model components. To search for such subspaces, we can adopt a natural generalization of full component activation patching which only patches the values of a subspace $U$ (while leaving the projection on its orthogonal complement $U^\perp$ unchanged). This was proposed in Geiger et al. (2023b), and closely related variants appear in Turner et al. (2023); Nanda et al. (2023); Lieberum et al. (2023).

For the purposes of exposition, we now restrict our discussion to activation patching of a 1-dimensional subspace (i.e. a *direction*) represented by a unit vector $v$. In this case, the subspace $U$ is the 1-dimensional subspace spanned by the vector $v^2$. If $\text{act}_A, \text{act}_B \in \mathbb{R}^d$ are the activations of a model component $\mathcal{C}$ on examples $A, B$ and $p_A = v^\top \text{act}_A, p_B = v^\top \text{act}_B$ are their projections along $v$, patching from $A$ into $B$ along $v$ results in the patched activation

$$\text{act}_B^{\text{patched}} = \text{act}_B + (p_A - p_B)v \tag{1}$$

**Intuition**. When will the update in equation 1 change the model's output in the intended way? Intuitively, two properties are necessary: $v$ must be activated differently by the two prompts (otherwise $p_A \approx p_B$ and the patch has no effect), and $v$ must be causally connected to the model's outputs (otherwise, if e.g. $v$ is in the nullspace of downstream model components, changing the activation along $v$ won't change model predictions). A direction $v$ faithful to the model's computation will simultaneously have these two properties.

The crux of the illusion is that $v$ may obtain each of the two properties from two unrelated directions in activation space, as shown in Figure 1. Specifically, we can form $v = v_{\text{disconnected}} + v_{\text{dormant}}$, where $v_{\text{disconnected}}$ distinguishes between the two prompts, but is in the nullspace of all downstream model components; and $v_{\text{dormant}}$ can *in principle* steer the model in the way intended by the patch, but is not activated differently by the two prompts. By patching along the sum of these directions, the variation in the disconnected part activates the dormant part, which then achieves the causal effect; this is illustrated in Figure 1[3].

**Formalization**. Let $\mathcal{M} : \mathcal{X} \to \mathcal{O}$ be a machine learning model that on input $x \in \mathcal{X}$ outputs a vector $y \in \mathcal{O}$ of probabilities over a set of output classes. Let $\mathcal{D}$ be a distribution over $\mathcal{X}$, and $\mathcal{C}$ be a component of $\mathcal{M}$, such that for $x \sim \mathcal{D}$ the hidden activation of $\mathcal{C}$ is a vector $c_x \in \mathbb{R}^d$. For a subspace $U_\mathcal{C} \subset \mathbb{R}^d$, we let $u_x$ be the orthogonal projection of $c_x$ onto $U_\mathcal{C}$. Finally, let $\mathcal{M}_{U_\mathcal{C} \leftarrow u_y}(x)$ be the result of running $\mathcal{M}$ with the input $x$ and setting the subspace $U_\mathcal{C}$ patched to $u_y$.

We say $U$ is *causally disconnected* if $\mathcal{M}_{U_\mathcal{C} \leftarrow u'}(x) = \mathcal{M}(x)$ for all $u' \in U$. In other words, setting the value of a causally disconnected subspace to any vector has no effect on model outputs. We say $U$ is *dormant* if $\mathcal{M}_{U_\mathcal{C} \leftarrow u_y}(x) \approx \mathcal{M}(x)$ with high probability over $x, y \sim \mathcal{D}$, but given $x \sim \mathcal{D}$, there exists $u$ such that $\mathcal{M}_{U_\mathcal{C} \leftarrow u}(x) \not\approx \mathcal{M}(x)$. In other words, a dormant subspace is approximately

---

[2]We remark that the illusion also applies to higher-dimensional subspaces (see Appendix A.1 for details).

[3]By contrast, patching along either of the two components individually has no effect: setting only the projection along $v_{\text{disconnected}}$ to any value has no effect on outputs by definition; and patching only along $v_{\text{dormant}}$ will have a weak effect because both examples activate it similarly.

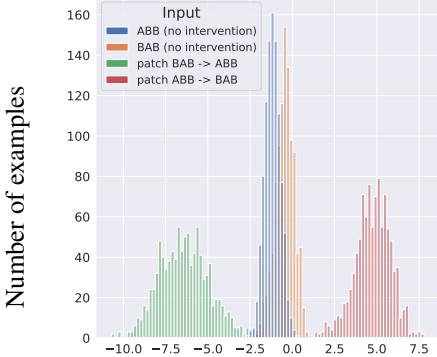

Figure 3: Projections of the output of the MLP layer on the gradient direction $\mathbf{v}_{\mathrm{grad}}$ before (blue/orange) and after (green/red) the activation patch along $\mathbf{v}_{\mathrm{MLP}}$. Here and elsewhere, 'ABB' denotes prompts where the **IO** name comes first, and 'BAB' denotes prompts where the **S** name comes first.

| Patching subspace | FLDD | Interchange accuracy |
|---|---|---|
| full MLP | -8% | 0.0% |
| $\mathbf{v}_{\mathrm{MLP}}$ | 46.7% | 4.2% |
| $\mathbf{v}_{\mathrm{MLP}}$ rowspace | 13.5% | 0.2% |
| $\mathbf{v}_{\mathrm{MLP}}$ nullspace | 0% | 0.0% |
| full residual stream | 123.6% | 54.8% |
| $\mathbf{v}_{\mathrm{resid}}$ | 140.7% | 74.8% |
| $\mathbf{v}_{\mathrm{resid}}$ rowspace | 127.5% | 63.1% |
| $\mathbf{v}_{\mathrm{resid}}$ nullspace | 13.9% | 0.4% |
| $\mathbf{v}_{\mathrm{grad}}$ | 111.5% | 45.1% |
| $\mathbf{v}_{\mathrm{grad}}$ rowspace | 106.47 | 40.6% |
| $\mathbf{v}_{\mathrm{grad}}$ nullspace | 2.2% | 0.0% |

Table 1: Effects of activation patching of full components and 1-dimensional subspaces on the IOI task: fractional logit diff (higher means less successful patch; 100% means no change) and interchange accuracy (fraction of predictions flipped; higher means more successful patch).

causally disconnected on the data distribution, but can have substantial causal effect if set to out of distribution values. We present a minimal concrete example of the illusion in the language of these concepts in Appendix A.3, where we also discuss a concrete hypothesis to test for the illusion.

# 4 THE ILLUSION IN THE INDIRECT OBJECT IDENTIFICATION TASK

## 4.1 PRELIMINARIES

In Wang et al. (2023), the authors analyze how the decoder-only transformer language model GPT-2 Small (Radford et al., 2019) performs the *indirect object identification* task. In this task, the model is required to complete sentences of the form 'When Mary and John went to the store, John gave a bottle of milk to' (with the intended completion in this case being ' Mary'). We refer to the repeated name (John) as **S**, the subject, and the non-repeated name (Mary) as **IO**, the indirect object. We use the same model GPT-2 Small, with a dataset that spans 216 names, 144 objects (e.g., 'bottle of milk') and 75 places (e.g., 'store') and three templates, split equally between a train and test set (see Appendix B for more details).

Wang et al. (2023) suggest the model uses the algorithm 'Find the two names in the sentence, detect the repeated name, and predict the non-repeated name' to do this task. In particular, they find a set of four heads in layers 7 and 8 – the **S-Inhibition heads** – that output the signal responsible for *not* predicting the repeated name. The dominant part of this signal is of the form 'Don't attend to the name in first/second position in the first sentence' depending on where the **S** name appears (see Appendix A in Wang et al. (2023) for details). This signal is added to the residual stream[4] at the last token position, and is then picked up by another class of heads in layers 9, 10 and 11 – the **Name Mover heads** – which incorporate it in their queries to shift attention to the **IO** name and copy it to the last token position, so that it can be predicted (Appendix Figure 21).

---

[4]We follow the conventions of Elhage et al. (2021) when describing internals of transformer models. The residual stream at layer $k$ is the sum of the output of all layers up to $k - 1$, and is the input into layer $k$.

## 4.2 Finding Subspaces Mediating Name Position Information

Given these findings, a natural next question is: how, precisely, do the S-Inhibition heads communicate the positional signal to the name mover heads? In particular, 'don't attend to the first/second name' is plausibly a binary feature represented by a 1-dimensional subspace. In this subsection, we present several methods to look for such a subspace.

**Gradient of name mover attention scores**. The three name mover heads identified in Wang et al. (2023) will attend to one of the names, and the model will predict whichever name is attended to. The position feature matters mechanistically by determining whether they attend to **IO** over **S**. So we can take the gradient of the difference of attention scores of these heads on the **S** and **IO** names. For a given IOI example, we expect this gradient will encode the position signal, and some of the name signal. By averaging gradients over the entire distribution, we expect the name signal to average out, and the position signal to be reinforced. Motivated by this, we compute a direction $\mathbf{v}_{\text{grad}}$ as follows. We take the average gradient of the difference of attention scores of the three name mover heads between the positions where the **S** and **IO** names are, average this over 2000 samples from our test distribution, and normalize the resulting vector to have unit $\ell_2$ norm.

**Distributed alignment search**. Instead of taking the gradient in specific attention heads informed by previous work, we can also directly optimize for a direction that mediates the position signal. This is the approach taken by DAS (Geiger et al., 2023b). In our context, DAS optimizes for an activation subspace which, when activation patched from prompt $B$ into prompt $A$, makes the model behave as if the relative position of the **IO** and **S** names in the sentence is as in prompt $B$. Specifically, we train DAS to maximize the difference between the logits of the name that should be predicted if this patch succeeded, and the other name. We find that relatively small sample sizes (e.g., 500) are sufficient. This approach is based purely on whether the model outputs the correct answer, and does not make any assumptions about the role of name mover heads. We let $\mathbf{v}_{\text{MLP}}$ and $\mathbf{v}_{\text{resid}}$ be 1-dimensional subspaces found by DAS in the layer 8 MLP activations and layer 8 residual stream output at the last token, respectively.

## 4.3 Demonstrating the Illusion for the $\mathbf{v}_{\text{MLP}}$ Direction

**Methodology**. In this section, we perform all patches between examples that only differ in the variable we want to localize in the model, i.e. the position of the **S** and **IO** names in the first sentence. That is, we patch from e.g. 'Then, Mary and John went to the store. John gave a book to' into 'Then, John and Mary went to the store. John gave a book to', and vice-versa. We report all metrics on a test set that is not used to train the interventions that require training. We consider the following activation patching interventions:

- **full MLP**: patching the hidden activation of the 8-th MLP layer at the last token.
- $\mathbf{v}_{\text{MLP}}$: patching along the direction $\mathbf{v}_{\text{MLP}}$ found in Subsection 4.2.
- $\mathbf{v}_{\text{MLP}}$ **nullspace**: patching along the causally disconnected component of $\mathbf{v}_{\text{MLP}}$. This is the orthogonal projection $\mathbf{v}_{\text{MLP}}^{\text{nullspace}}$ of $\mathbf{v}_{\text{MLP}}$ on the nullspace $\ker W_{out}$ of the down-projection $W_{out}$ of the MLP layer.
- $\mathbf{v}_{\text{MLP}}$ **rowspace**: patching along the causally relevant component of $\mathbf{v}_{\text{MLP}}$. This is the orthogonal projection $\mathbf{v}_{\text{MLP}}^{\text{rowspace}}$ of $\mathbf{v}_{\text{MLP}}$ on the rowspace of $W_{out}$. Note that we have the orthogonal decomposition $\mathbf{v}_{\text{MLP}} = \mathbf{v}_{\text{MLP}}^{\text{nullspace}} + \mathbf{v}_{\text{MLP}}^{\text{rowspace}}$.
- **full residual stream**: patching the entire activation of the residual stream at the last token after layer 8 of the model. This is indicated as the location of $\mathbf{v}_{\text{resid}}$ in Figure 21.

All activation patches have the goal of making the model output the **S** name instead of the **IO** name. Accordingly, we use the *logit difference* between the logits assigned to the **IO** and **S** names to measure how well a patch performs. Formally, let $\text{logit}_{\text{IO}}(x), \text{logit}_{\text{S}}(x)$ denote the last-token logits output by the model for the **IO** and **S** names respectively on input $x$. The logit difference $\text{logitdiff}(x) := \text{logit}_{\text{IO}}(x) - \text{logit}_{\text{S}}(x)$ measures the confidence of the model for the **IO** name over the **S** name (it is the log-odds between the two names assigned by the model). It averages $\approx 3.3$ over the IOI distribution, and is positive for almost all examples (99%+). Similarly, for an activation patching intervention $P$, let $\text{logit}_{\text{IO}}^{P}(x), \text{logit}_{\text{S}}^{P}(x)$ be the corresponding logits when

run on $x$ with $P$ applied, with $\mathrm{logitdiff}^P(x) := \mathrm{logit}_{\mathbf{IO}}^P(x) - \mathrm{logit}_{\mathbf{S}}^P(x)$. Our main metric is the average **fractional logit difference decrease (FLDD)** due to a patching intervention $P$, given by $\mathrm{FLDD}^P(x) = 1 - \frac{\mathrm{logitdiff}^P(x)}{\mathrm{logitdiff}(x)}$. This metric is zero when the patch has no effect on the logit difference on average, and values above 100% suggest that the patch more often than not makes the model prefer the **S** name over the **IO** name. We also report the **interchange accuracy**, which is the fraction of prompts for which the **IO** name is assigned a higher logit than the **S** name, but the intervention $P$ reverses this.

**Results**. Metrics are shown in Table 1. Through these metrics and additional experiments, we exhaustively confirm the mechanics of the illusion.

THE CAUSALLY DISCONNECTED COMPONENT OF $\mathbf{v}_{\mathrm{MLP}}$ DRIVES THE EFFECT. While patching the $\mathbf{v}_{\mathrm{MLP}}$ direction has a significant effect on the FLDD metric (46.7%), this effect is greatly diminished when we remove the component of $\mathbf{v}_{\mathrm{MLP}}$ in $\ker W_{out}$ whose activations are (provably) causally disconnected from model predictions (13.5%), or when we patch the entire MLP activation (−8%, actually increasing confidence). By contrast, performing analogous ablations on $\mathbf{v}_{\mathrm{resid}}$ leads to very similar numbers (140.7%/127.5%/123.6%; we refer the reader to Section 5 for details on the $\mathbf{v}_{\mathrm{resid}}$ experiments). This confirms our hypothesis from Section 4.

PATCHING $\mathbf{v}_{\mathrm{MLP}}$ ACTIVATES A DORMANT PATHWAY THROUGH THE MLP. To corroborate these findings, in Figure 3, we plot the projection of the MLP layer's contribution to the residual stream on the gradient direction $\mathbf{v}_{\mathrm{grad}}$ before and after patching, in order to see how it contributes to the attention of name mover heads. We observe that in the absence of intervention, the MLP output is weakly sensitive to the name position information, whereas after the patch this changes significantly.

FURTHER VALIDATIONS OF THE ILLUSION. We observe that the disconnected-dormant decomposition from the illusion approximately holds: the causally disconnected component of $\mathbf{v}_{\mathrm{MLP}}$ (the one in $\ker W_{out}$) is significantly more activated by the position information than the component in $(\ker W_{out})^{\perp}$, which is the one driving the causal effect (Appendix Figure 20); in this sense, the causally relevant component is 'dormant' relative to the causally diconnected one. While the contribution of the $\mathbf{v}_{\mathrm{MLP}}$ patch to logit difference may appear relatively small, in Appendix B.4 we argue that this is significant for a single component. Finally, in Appendix B.5, we show that we can find a direction within the post-$\mathrm{gelu}$ activations that has an even stronger effect on the model's behavior, *even when* we replace the MLP weights with random matrices.

## 5 FINDING AND VALIDATING A FAITHFUL DIRECTION MEDIATING NAME POSITION IN THE IOI TASK

As a counterpoint to the illusion, in this section we demonstrate a success case for subspace activation patching and DAS by revisiting the directions $\mathbf{v}_{\mathrm{grad}}$ and $\mathbf{v}_{\mathrm{resid}}$ defined in Subsection 4.2, and arguing they are faithful to the model's computation to a high degree. Specifically, we subject these directions to the same tests we used for the illusory direction $\mathbf{v}_{\mathrm{MLP}}$ and arrive at significantly different results. Through this and additional validations, we demonstrate that these directions possess the necessary and sufficient properties of a successful activation patch – being both correlated with input variation and causal for the targeted behavior – in an irreducible way.

**Ruling out the illusion**. Let $W_Q^{\mathrm{name\ movers}} \in \mathbb{R}^{768 \times 192}$ be the stacked query matrices of the name mover heads. We use this matrix as a proxy to determine the causally disconnected subspace of the 768-dimensional residual stream. In Table 1, we show the fractional logit difference and interchange accuracy when patching $\mathbf{v}_{\mathrm{resid}}$ and $\mathbf{v}_{\mathrm{grad}}$, as well as their components along $\ker W_Q^{\mathrm{name\ movers}}$ (denoted 'nullspace') and its orthogonal complement (denoted 'rowspace'). We observe that the non-nullspace metrics are broadly similar; in particular, removing the causally disconnected component of $\mathbf{v}_{\mathrm{resid}}$ does not greatly diminish the effect of the patch in terms of the logit difference metrics (as it does for $\mathbf{v}_{\mathrm{MLP}}$). We also find that $\mathbf{v}_{\mathrm{resid}}$ is predominantly in $(\ker W_Q^{\mathrm{name\ movers}})^{\perp}$ (and so is $\mathbf{v}_{\mathrm{grad}}$, but this is to be expected by its definition).

Importantly, since the residual stream activation where $\mathbf{v}_{\mathrm{resid}}$ and $\mathbf{v}_{\mathrm{grad}}$ are patched is a full bottleneck for the model's computation, it is not possible for these directions to be causal but dormant (in the sense of Section 3): there can be no earlier model component that activates this direction in a way that avoids the patch via a skip connection (unlike for the $\mathbf{v}_{\mathrm{MLP}}$ direction). Indeed, in Figure 7 in Appendix C we show that the $\mathbf{v}_{\mathrm{resid}}$ direction gets written to by the S-Inhibition heads, and in Figure

19 in Appendix F.2, we show they strongly discriminate between the prompts where the **IO** name comes first/second. See C for additional experiments validating these subspaces.

## 6 FACTUAL RECALL

Given a fact expressed as a subject-relation-object triple $(s, r, o)$ (e.g., $s = $ 'Eiffel Tower', $r = $ 'is in', $o = $ 'Paris'), we say that a model $M$ 'recalls' the fact $(s, r, o)$ if $M$ completes a prompt expressing just the $(s, r)$ pair (e.g., 'The Eiffel Tower is in') with $o$. In this section, we show that the interpretability illusion can also be exhibited for the factual recall capability of language models, a much broader setting than the IOI task. In particular, we can fit DAS to any such pair of facts and perform an activation patch that changes recall. We further show that the illusory subspace is equivalent to a rank one edit to the weights that changed the recalled fact.

### 6.1 FINDING ILLUSORY 1-DIMENSIONAL PATCHES FOR FACTUAL RECALL

Let us be given two facts $(s, r, o)$ and $(s', r, o')$ for the same relation that a model recalls correctly, with corresponding factual prompts $A$ expressing $(s, r)$ and $B$ expressing $(s', r)$ (e.g., $r = $ 'is in', $A = $ 'The Eiffel Tower is in', $B = $ 'The Colosseum is in'). In this subsection, we patch from $B$ into $A$, with the goal of changing the model's output from $o$ to $o'$.

More precisely, we patch from the last token of $s'$ in $B$ to the last token of $s$ in $A$ (prior work has shown that the fact is retrieved on $s$ (Geva et al., 2023) ), and we again use DAS Geiger et al. (2023b) to optimize for a direction that maximizes the logit difference between $o'$ and $o$. We use the first 1000 examples from the COUNTERFACT dataset (Meng et al., 2022a), and filter from them 40 pairs of facts across 8 relations that the model recalls correctly. We use GPT-2 XL (1.5B parameters) for experiments.

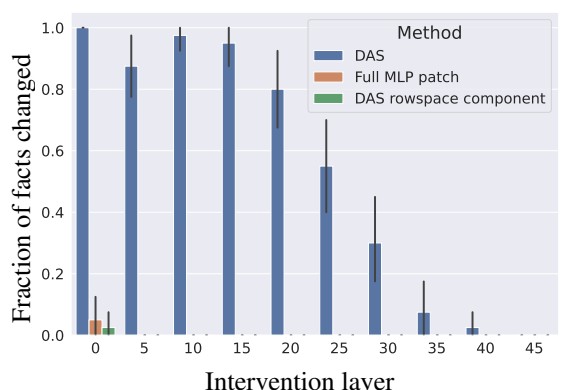

Figure 4: Fraction of successful fact patches under three interventions: patching along the direction found by DAS (blue), patching the component of the DAS direction in the rowspace of $W_{out}$ (green), and patching the entire hidden MLP activation (orange).

Similar to Section 4, we compare the effect of these patches to a full MLP patch, and a patch of only the rowspace component. Results are shown in figure 4. We find a stronger version of the same qualitative phenomena we observed with the IOI illusory direction: (i) the directions we find have a strong causal effect (successfully changing $o$ to $o'$), but (ii) this effect disappears when we ablate the component in the nullspace of $W_{out}$, and (iii) patching the entire MLP activation instead has a negligible effect on the difference in logits between the correct and incorrect objects. We further observe that the difference in last-token subject activations is significantly aligned with the causally disconnected component of the subspaces found (cosine similarity $\sim 0.9$, Appendix Figure 12). This implies that it is weakly aligned with the rowspace component (cosine similarity $\sim 0.4$); hence, the rowspace component is relatively dormant compared to the causally disconnected component. Further experiments confirming the illusion are in Appendix D.2.

### 6.2 1-DIMENSIONAL FACT PATCHES IMPLY EQUIVALENT RANK-1 FACT EDITS

Finally, we show that the existence of an activation patch as in Subsection 6.1 implies the existence of a seemingly different intervention with similar effect: a *rank-one model edit* to the weights of the MLP layer. Proposed in Meng et al. (2022a), rank-1 editing modifies the $W_{out}$ weight of a single MLP layer to make a model that recalls $(s, r, o)$ recall $(s, r, o')$ instead, while minimally modifying the model otherwise. This finding suggests a mechanistic explanation for the discrepancy between fact localization and fact editing observed in prior work (Hase et al., 2023). Namely, illusory subspaces where fact editing can 'work' may exist even in layers not storing the fact, by virtue of the

residual stream containing a causally disconnected but sensitive to the subject direction and a causal for the object direction to be combined in an MLP layer.

In more detail, Meng et al. (2022a) propose a specific rank-1 model edit, ROME, which takes as input a vector $k \in \mathbb{R}^{d_{MLP}}$ representing the subject (e.g. an average of last-token representations of the subject) and a vector $v \in \mathbb{R}^{d_{model}}$ which, when output by the MLP layer, will cause the model to predict a new object for the factual prompt. ROME modifies the MLP weight by setting $W'_{out} = W_{out} + ab^{\top}$, where $a \in \mathbb{R}^{d_{model}}, b \in \mathbb{R}^{d_{MLP}}$ are chosen so that $W'_{out}k = v$, and the MLP's output is otherwise minimally changed. Without loss of generality, the first condition implies that $a = v - W_{out}k$ and $b^{\top}k = 1$; the second condition is then modeled by minimizing the variance of $b^{\top}x$ when $x \sim \mathcal{N}(0, \Sigma)$ for an empirical estimate $\Sigma \in \mathbb{R}^{d_{MLP} \times d_{MLP}}$ of the covariance of MLP activations (see Lemma D.1 in Appendix D for details and a proof).

Intuitively, a fact patch as in subsection 6.1 should have a corresponding ROME edit with the same effect. Specifically, suppose that we are in the setup of 6.1, and $u_A, u_B$ are the last subject token post-GELU activations for prompts $A$ and $B$; then $u_A$ takes the role of $k$, and the patched output of the MLP layer takes the role of $v$. Indeed, in Appendix D.4 we show that, for any direction $v \in \mathbb{R}^{d_{MLP}}$ in the MLP's activation space, there exists a rank-1 edit $W'_{out} = W_{out} + ab^{\top}$ which results in the same output for the MLP layer at the last subject token of $A$ as activation-patching from $u_B$ into $u_A$ along $v$; the formal statement and proof are given in D.4, where we also derive how to make such an edit minimally change the model, following the optimization objective of ROME.

While this shows that the patch implies a rank-1 edit with the same behavior *at the token being patched*, the rank-1 edit is applied *permanently* to the model, which means that it (unlike the activation patch) applies to *every* token. Thus, it is not a priori obvious whether the rank-1 edit will still succeed in making the model predict $o'$ instead of $o$. To this end, in Appendix D.2, we evaluate empirically how using the rank-1 edit from Lemma D.2 instead of the activation patch changes model predictions, and we find negligible differences.

## 7 DISCUSSION AND CONCLUSION

**Do we expect this illusion to be prevalent?** We only exhibit our illusion empirically in two settings, IOI and factual recall, but we believe it is likely prevalent in practice. We do not prove this, but hope to illustrate it with an informal argument. Specifically, we expect the illusion to occur whenever we have an MLP $M$ which is not used in the model's computation on a given task, but is between two components $A$ and $B$ which are used and compose with each other via some direction $v$ in the residual stream (i.e., $v$ is communicated via the skip connections of intervening layers). This is a common structure that has been frequently observed in the mechanistic interpretability literature (Lieberum et al., 2023; Wang et al., 2023; Olsson et al., 2022; Geva et al., 2021): circuits contain components composing with each other separated by multiple layers, and circuits have often been observed to be sparse, with most components (including most MLP layers) not playing a significant role.

**An empirically informed mechanistic argument for the illusion**. The MLP $M$ between $A$ and $B$ (MLP-in-the-middle) set up leads to the existence of both a dormant causal direction $v_{dormant}$ and a correlated yet causally disconnected direction $v_{disconnected}$, as follows. A vector $u \in \mathbb{R}^{d_{MLP}}$ in the MLP's activation space will be causally relevant if $W_{out}u = v$; since the internal dimension of the MLP layer is larger than that of the residual stream ($4\times$ for popular architectures), $W_{out}$ is typically full rank (see Appendix E.1 for empirical evidence), and such a $u$ will exist. In particular, we can take $v_{dormant} = W_{out}^{+}v$, and since we assume $M$ is not a part of the computation, this direction should not be significantly correlated with the feature being patched. A vector $u \in \mathbb{R}^{d_{MLP}}$ will be correlated if it can linearly recover the concept in question from the MLP's activations $\text{gelu}(W_{in}x_{resid})$. Because $M$ occurs after $A$, the feature is linearly recoverable from $x_{resid}$ by using $v$ as a probe, and hence is recoverable from $W_{in}x_{resid}$ as well (again, empirically $W_{in}$ is full-rank). In theory, the $\text{gelu}$ non-linearity could completely destroy the information, but empirically (see Appendix E.1 and E.2), it seems to merely add noise, so the feature remains linearly recoverable, giving us $v_{disconnected}$.

**Takeaways and recommendations**. As we have seen, optimization-based methods using subspace activation patching can find both faithful and illusory features with respect to the model's computation. We recommend running such methods in activation bottlenecks such as the residual stream, as well as using validations beyond end-to-end evaluation to ascertain the precise role of such features.

ACKNOWLEDGEMENTS

We would like to thank the Stanford Existential Risk Initiative's Machine Learning Alignment and Theory Scholars (SERI MATS) program (Summer 2023 Cohort) for supporting this research, and to Atticus Geiger for many helpful and insightful discussions. We would also like to thank the anonymous reviewers for their helpful feedback.

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

# A    ADDITIONAL DETAILS FOR SECTION 3

## A.1    THE ILLUSION FOR HIGHER-DIMENSIONAL SUBSPACES

In the main text, we mostly discuss the illusion for activation patching of 1-dimensional subspaces for ease of exposition. Here, we develop a more complete picture of the mechanics of the illusion for higher-dimensional subspaces.

Let $\mathcal{C}$ be a model component taking values in $\mathbb{R}^d$, and let $U \subset \mathbb{R}^d$ be a linear subspace. Let $V$ be a matrix whose columns form an orthonormal basis for $U$. If the $\mathcal{C}$ activations for examples $A$ and $B$ are $\text{act}_A, \text{act}_B \in \mathbb{R}^d$ respectively, patching $U$ from $A$ into $B$ gives the patched activation

$$\text{act}_B^{patched} = \text{act}_B + VV^\top(\text{act}_A - \text{act}_B) = (I - VV^\top)\text{act}_B + VV^\top\text{act}_A$$

For intuition, note that $VV^\top$ is the orthogonal projection on $U$, so this formula says to replace the orthogonal projection of $\text{act}_B$ on $U$ with that of $\text{act}_A$, and keep the rest of $\text{act}_B$ the same.

Generalizing the discussion from Section 3, for the illusion to occur for subspace $S$, we need $S$ to be sufficiently aligned with a causally disconnected subspace $V_{disconnected}$ that is correlated with the feature being patched, and a dormant but causal subspace $V_{dormant}$ which, when set to out of distribution values, can achieve the wanted causal effect. For example, a particularly simple way in which this could happen is if we have let $V_{disconnected}, V_{dormant}$ be 1-dimensional subspaces (like in the setup for the 1-dimensional illusion), and we form $S$ by combining $V_{disconnected} + V_{dormant}$ with a number of orthogonal directions that are approximately constant on the data with respect to the feature we are patching. These extra directions effectively don't matter for the patch (because they are cancelled by the $\text{act}_A - \text{act}_B$ term). Given a specific feature, it is likely that such weakly-activating directions will exist in a high-dimensional activation space. Thus, if the 1-dimensional illusion exist, so will higher-dimensional ones.

## A.2    ILLUSORY 1-DIMENSIONAL PATCHES ARE APPROXIMATELY EQUAL PARTS CAUSALLY DISCONNECTED AND DORMANT

In this subsection, we prove a quantitative corollary of the model of our illusion that suggests that we should expect illusory patching directions to be of the form $v = \frac{1}{\sqrt{2}}\left(v_{disconnected} + v_{dormant}\right)$ for unit vectors $\|v_{disconnected}\|_2 = \|v_{dormant}\|_2 = 1$. In other words, we expect the best illusory patches to be formed by combining a disconnected and illusory direction with equal coefficients, like depicted in Figure 1:

**Lemma A.1.** *Suppose we have two distributions of input prompts $\mathcal{D}_A, \mathcal{D}_B$. In the terminology of Section 3, let $v_{disconnected} \perp v_{dormant}$ be unit vectors such that the subspace spanned by $v_{disconnected}$ is a causally disconnected subspace, and the subspace spanned by $v_{dormant}$ is strongly dormant, in the sense that the projections of the activations of all examples $\mathcal{D}_{source} \cup \mathcal{D}_{base}$ onto $v_{dormant}$ are equal to some constant c.*

*Suppose we form $v = v_{disconnected}\cos\alpha + v_{dormant}\sin\alpha$ as a unit-norm linear combination of the two directions. Then the magnitude of the expected change in projection along $v_{dormant}$ when patching from $x_A \sim \mathcal{D}_A$ into $x_B \sim \mathcal{D}_B$ is maximized when $\alpha = \frac{\pi}{4}$, i.e. $\cos\alpha = \sin\alpha = \frac{1}{\sqrt{2}}$.*

*Proof.* Recall that the patched activation from $x_A$ into $x_B$ along $v$ is

$$\text{act}_B^{patched} = \text{act}_B + (p_A - p_B)v \tag{2}$$

where $p_A = v^\top\text{act}_A, p_B = v^\top\text{act}_B$ are the projections of the two examples' activations on $v$. The change along $v_{dormant}$ is thus

$$v_{dormant}^\top\left(\text{act}_B^{patched} - \text{act}_B\right) = (p_A - p_B)\sin\alpha = (v^\top\text{act}_A - v^\top\text{act}_B)\sin\alpha$$

$$= v_{disconnected}^\top(\text{act}_A - \text{act}_B)\cos\alpha\sin\alpha$$

where we used the assumption that $v_{dormant}^\top\text{act}_A = v_{disconnected}^\top\text{act}_B$. Hence, the expected change is

$$\cos\alpha\sin\alpha\, v_{disconnected}^\top\mathbb{E}_{A\sim\mathcal{D}_A, B\sim\mathcal{D}_B}\left[\text{act}_A - \text{act}_B\right].$$

The function $f(\alpha) = \cos\alpha\sin\alpha$ for $\alpha \in [0, \pi/2]$ is maximized for $\alpha = \pi/4$, concluding the proof.  □

### A.3 CONCRETE MATHEMATICAL EXAMPLE OF THE ILLUSION

**A Toy Illusion**. For a distilled example of the illusion, consider a network $\mathcal{A}$ that takes in a real valued input $x \in \mathbb{R}$, computes a three dimensional hidden representation $h = W_1^T x$, and then a real valued output $y = W_2^T h$. In this example, we consider the network's hidden layer to be a single 'component', represented as a 3-dimensional vector space of activations. Define the weights to be $W_1 = [1 \quad 0 \quad 1]$ and $W_2 = [0 \quad -2 \quad 1]$ and observe that the network computes the identity function (see Figure 18 in Appendix F.1 for an illustration). While the 3rd hidden neuron clearly mediates this effect, surprisingly, patching the direction along the sum of the first two neurons does as well, despite the fact that the 1st neuron is causally disconnected, and the 2nd is dormant.

Consider Figure 18. It should be obvious that the hidden unit $H_3$ fully mediates the information flow from input to output, and that $H_2$ is dormant while $H_3$ is disconnected. However, it may be surprising that the linear subspace of $H_1$ and $H_2$ defined by the unit vector $[\frac{1}{\sqrt{2}}, -\frac{1}{\sqrt{2}}]$ also fully mediates the information flow, despite it consisting of dormant and disconnected directions. Activation patching on this subspace leverages the information stored in the disconnected subspace in order to activate the dormant subspace by fixing it to an out of distribution value. In this way, activation patching on a subspace can activate a 'dormant parallel circuit'.

**A concrete hypothesis**. Based on this discussion, we can also postulate a specific hypothesis we can test for. Specifically, our hypothesis is that there exist pre-trained transformer language models $M$, pairs of distributions $\mathcal{D}_{base}$ and $\mathcal{D}_{source}$ over inputs to $M$ (with associated ground-truth next-word predictions for inputs in $D_{base} \cup D_{source}$), and 1-dimensional subspaces $S$ of post-GELU activations of MLP layers in these models, such that activation patching from $x_{source} \sim D_{source}$ into $x_{base} \sim D_{base}$ along $S$ has a strong effect of shifting probability from the expected completion of $x_{base}$ to that of $x_{source}$, but this effect is significantly diminished when activation patching is performed along the component of $S$ orthogonal to the nullspace of the down-projection $W_{out}$ of the MLP layer. In this example, the nullspace component of $S$ is the causally disconnected subspace, and we hypothesise it is possible the remaining component is dormant.

# B ADDITIONAL DETAILS FOR SECTION 4

### B.1 DATASET, MODEL AND EVALUATION DETAILS FOR THE IOI TASK

We use GPT2-Small for the IOI task, with a dataset that spans 216 single-token names, 144 single-token objects and 75 single-token places, which are split $1 : 1$ across a training and test set. Every example in the data distribution includes (i) an initial clause introducing the indirect object (**IO**, here 'Mary') and the subject (**S**, here 'John'), and (ii) a main clause that refers to the subject a second time. Beyond that, the dataset varies in the two names, the initial clause content, and the main clause content. Specifically, use three templates as shown below:

> Then, [ ] and [ ] had a long and really crazy argument. Afterwards, [ ] said to
> Then, [ ] and [ ] had lots of fun at the [place]. Afterwards, [ ] gave a [object] to
> Then, [ ] and [ ] were working at the [place]. [ ] decided to give a [object] to

and we use the first two in training and the last in the test set. Thus, the test set relies on unseen templates, names, objects and places. We used fewer templates than the IOI paper Wang et al. (2020) in order to simplify tokenization (so that the token positions of our names always align), but our results also hold with shifted templates like in the IOI paper.

On the test partition of this dataset, GPT2-Small achieves an accuracy of $\approx 91\%$. The average difference of logits between the correct and incorrect name is $\approx 3.3$, and the logit of the correct name is greater than that of the incorrect name in $\approx 99\%$ of examples. Note that, while the logit difference is closely related to the model's correctness, it being $> 0$ does not imply that the model makes the correct prediction, because there could be a third token with a greater logit than both names.

## B.2 Details for Computing the Gradient Direction $\mathbf{v}_{\text{GRAD}}$

For a given example from the test distribution and a given name mover head, we compute the gradient of the difference of attention scores from the final token position to the 3rd and 5th token in the sentence (where the two name tokens always are in our data). We then average these gradients over a large sample of the full test distribution and over the three name mover heads, and finally normalize the resulting vector to have unit $\ell_2$ norm.

We note that there is a 'closed form' way to compute approximately the same quantity that requires no optimization. Namely, for a single example we can collect the keys $k_S, k_{IO}$ to the name mover heads at the first two names in the sentence (the **S** and **IO** name). Then, for a single name mover head with query matrix $W_Q$, a maximally causal direction $v$ in the residual stream at the last token position after layer 8 will be one such that $W_Q v$ is in the direction of $k_S - k_{IO}$, because the attention score is simply the dot product between the keys and queries. We can use this to 'backpropagate' to $v$ by multiplying with the pseudoinverse $W_Q^+$. This is slightly complicated by the fact that we have been ignoring layer normalization, which can be approximately accounted for by estimating the scaling parameters (which tend to concentrate well) from the IOI data distribution. We note that this approach leads to broadly similar results.

## B.3 Training Details for DAS

To train DAS, we always sample examples from the training IOI distribution as described in Appendix B. We sample equal amounts of pairs of base (which will be patched into) and source (where we take the activation to patch in from) prompts where the two names are the same between the prompts, and pairs of prompts where all four names are distinct. We optimize DAS to maximize the logit difference between the name that should be predicted if the position information from the source example is correct and the other name.

For training, we use a learned rotation matrix as in the original DAS paper (Geiger et al., 2023b), parametrized with `torch.nn.utils.parametrizations.orthogonal`. We use the Adam optimizer and minibatch training over a training set of several hundred patching pairs. We note that results remain essentially the same when using a higher number of training examples.

## B.4 Discussion of the Magnitude of the Illusion

While the contribution of the $\mathbf{v}_{\text{MLP}}$ patch to logit difference may appear relatively small, we note that this is the result of patching a direction in a single model component at a single token position. Typical circuits found in real models (including the IOI circuit from Wang et al. (2023)) are often composed of multiple model components, each of which contribute. In particular, the position signal itself is written to by 4 heads, and chiefly read by 3 other heads. As computation tends to be distributed, when patching an individual component accuracy may be a misleading metric (eg patching 1 out of 3 heads is likely insufficient to change the output), and a fractional logit diff indicates a significant contribution. By contrast, patching in the residual stream is a more potent intervention, because it can affect *all* information accumulated in the model that is communicated to downstream components.

## B.5 Random ablation of MLP weights

How certain are we that MLP8 doesn't actually matter for the IOI task? While we find the IOI paper analysis convincing, to make our results more robust to the possibility that it does matter, we also design a further experiment.

Given our conceptual picture of the illusion, the computation performed by the MLP layer where we find the illusory subspace does not matter as long as it propagates the correlational information about the position feature from the residual stream to the hidden activations, and as long as the output matrix $W_{out}$ is full rank (also, see the discussion in 7). Thus, we expect that if we replace the MLP weights by randomly chosen ones with the same statistics, we should still be able to exhibit the illusion.

Specifically, we randomly sampled MLP weights and biases such that the norm of the output activations matches those of MLP8. As random MLPs might lead to nonsensical text generation, we don't replace the layer with the random weights, but rather train a subspace using DAS on the MLP activations, and add the difference between the patched and unpatched output of the random MLP to the real output of MLP8. This setup finds a subspace that reduces logit difference even more than the $\mathbf{v}_{\text{MLP}}$ direction.

This suggests that the existence of the $\mathbf{v}_{\text{MLP}}$ subspace is less about *what* information MLP8 contains, and more about *where* MLP8 is in the network.

## C  ADDITIONAL DETAILS FOR SECTION 5

**Which model components write to the $\mathbf{v}_{\text{resid}}$ direction?** To test how every attention head and MLP contributes to the value of projections on $\mathbf{v}_{\text{MLP}}$, we sampled activations from head and MLP outputs at the last token position of IOI prompts, and calculated their dot product with $\mathbf{v}_{resid}$ (Figure 7). We found that the dot products of most heads and MLPs was low, and that the S-inhibition heads were the only heads whose dot product differed between different patterns ABB and BAB. This shows that only the S-inhibition heads write to the $\mathbf{v}_{\text{resid}}$ direction (as one would hope). Importantly, this test separates $\mathbf{v}_{resid}$ from the interpretability illusion $\mathbf{v}_{MLP}$. While patching $\mathbf{v}_{\text{MLP8}}$ also writes to $\mathbf{v}_{\text{resid8}}$ (i.e. $\mathbf{v}_{\text{MLP8}}W_{out} \approx \mathbf{v}_{\text{resid8}}$), the MLP layer does not write this subspace on the IOI task (see Figure 3). This further supports the observation that the $\mathbf{v}_{\text{MLP}}$ patch activates a dormant pathway in the model.

**Generalization beyond the IOI distribution**. We also investigate how the subspace generalizes. We sample prompts from OpenWebText-10k and look at those with particularly high and low activations in $\mathbf{v}_{\text{sinhib}}$. Representative examples are shown in Figure 8 together with the name movers attention at the position of interest, how the probability changes after subspace ablation, and how the name movers attention changes.

**Stability of found solution**. Finally, we note that solutions found by DAS in the residual stream are stable, including when trained on a subset of S-inhibition heads (see Figure 5).

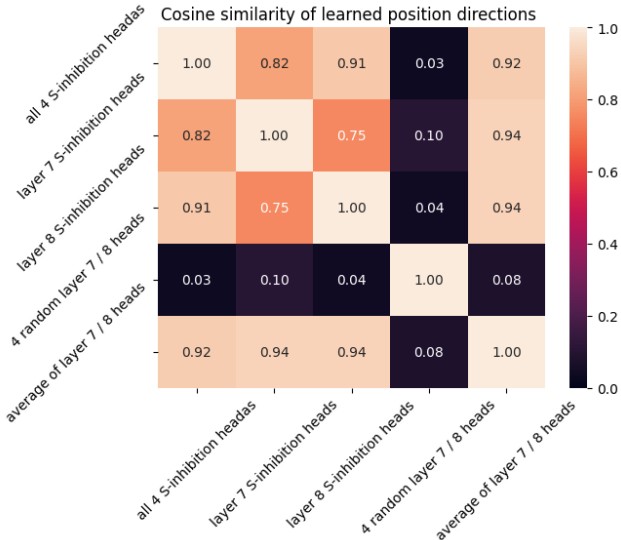

Figure 5: Cosine Similarity between learned position subspaces in the S-inhibition heads is high even when using only a subset of S-inhibition heads for training

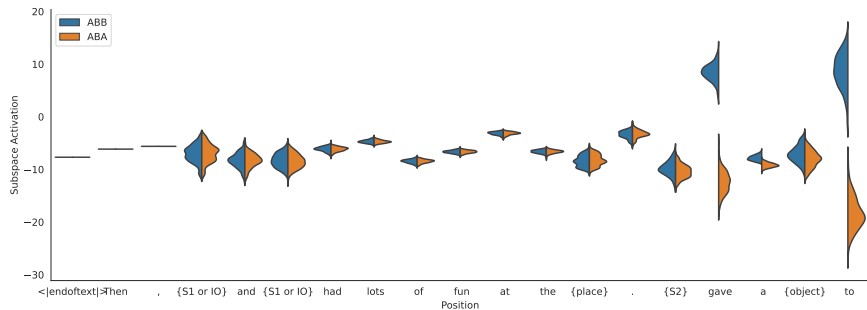

Figure 6: The IOI position subspace activates at words that predict a repeated name. S-inhibition subspace activations for different IOI prompts per position

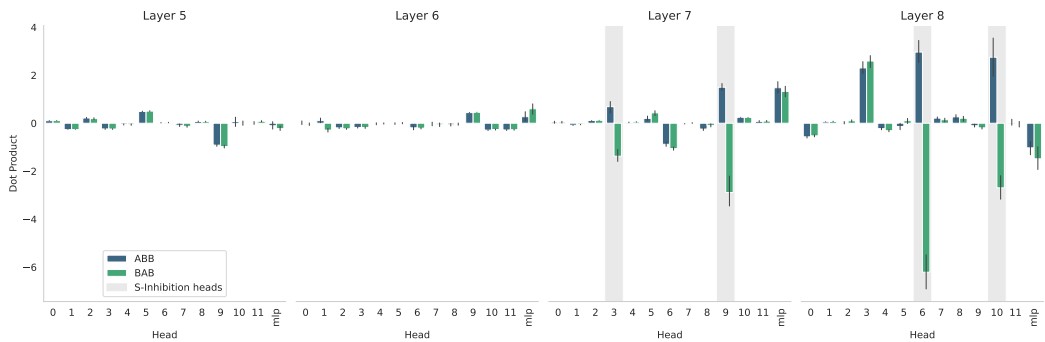

Figure 7: S-Inhibition heads but not MLP8 write to the position subspace in the residual stream that is causally connected to the name movers on the IOI task

# D  ADDITIONAL DETAILS FOR SECTION 6

## D.1  TRAINING DETAILS FOR FACT PATCHING (SECTION 6.1)

We use the first 1000 examples from the COUNTERFACT dataset (Meng et al., 2022a). We filter the facts which GPT2-XL correctly recalls. Out of the remaining facts, for each relation we form all pairs of distinct facts, and we sample 5 such pairs from each relation with at least 5 facts. This results in a collection of 40 facts spanning 8 different relations.

To train DAS, due to the computational difficulty of the problem, instead of optimizing a rotation matrix like in Geiger et al. (2023b), we directly optimize for a single unit vector using projected gradient descent, where after each gradient update we normalize the vector to have unit norm again.

## D.2  ADDITIONAL FACT PATCHING EXPERIMENTS

In figure 11, we show the distribution of the fractional logit difference metric (see Subsection 4.2 for a definition) when patching between facts as described in Subsection 6.1. Like in the related Figure 4, we observe that, while patching along the directions found by DAS achieves strongly negative values (indicating that the facts are very often successfully changed by the patch), the interventions that replace the entire MLP layer or only the causally relevant component of the DAS directions have no such effect.

Next, we observe that the nullspace component of the patching direction is the one similar to the variation in the inputs (difference of last-token activations at the two subjects). Specifically, in Figure 12, we plot the (absolute value of the) cosine similarity between the difference in activations for the two last subject tokens, and the nullspace component of the DAS direction. We note that

this similarity is consistently significantly high (note that it can be at most 1, which would indicate perfect alignment).

Finally, we observe that the nullspace component of the patching direction is a non-trivial part of the direction in Figure 13, where we plot the distribution of the $\ell_2$ norm of this component.

### D.3 RANK-1 MODEL EDITING AS AN OPTIMIZATION PROBLEM

We now review the ROME method from Meng et al. (2022a) and show how it can be characterized as the solution of a simple optimization problem. Following the terminology of 6.2, let us have an MLP layer with an output projection $W_{out}$, a key vector $k \in \mathbb{R}^{d_{MLP}}$ and a value vector $v \in \mathbb{R}^{d_{model}}$.

In Meng et al. (2022a), equation 2, the formula for the rank-1 update to $W_{out}$ is given by

$$W'_{out} = W_{out} + (v - W_{out}k)\frac{k^\top \Sigma^{-1}}{k^\top \Sigma^{-1} k} \tag{3}$$

where $\Sigma$ is an empirical estimate of the uncentered covariance of the pre-$W_{out}$ activations. We derive the following equivalent characterization of this solution (which may be of independent interest):

**Lemma D.1.** *Given a matrix $W_{out} \in \mathbb{R}^{d_{model} \times d_{MLP}}$, a key vector $k \in \mathbb{R}^{d_{MLP}}$ and a value vector $v \in \mathbb{R}^{d_{model}}$, let $\Sigma \succ 0, \Sigma \in \mathbb{R}^{d_{MLP} \times d_{MLP}}$ be a positive definite matrix (specifically, the uncentered empirical covariance), and let $x \sim \mathcal{N}(0, \Sigma)$ be a normally distributed random vector with zero mean and covariance $\Sigma$. Then, the ROME weight update is $W'_{out} = W_{out} + ab^\top$ where $a \in \mathbb{R}^{d_{model}}, b \in \mathbb{R}^{d_{MLP}}$ solve the optimization problem*

$$\min_{a,b} \text{trace}(\text{Cov}_x \left[ W'_{out}x - W_{out}x \right]) \quad \text{subject to} \quad W'_{out}k = v.$$

*In other words, the ROME update is the update that causes $W_{out}$ to output $v$ on input $k$, and minimizes the total variance of the extra contribution of the update in the output of the MLP layer under the assumption that the pre-$W_{out}$ activations are normally distributed with covariance $\Sigma$[5].*

*Proof.* Using $\mathbb{E}_x[xx^\top] = \Sigma$ and the cyclic property of the trace, we see that

$$\text{trace}(\text{Cov}_x \left[ W'_{out}x - W_{out}x \right]) = \|a\|_2^2 b^\top \Sigma b$$

We must have $ab^\top k = v - Wk$, so without loss of generality we can rescale $a, b$ so that $a = v - Wk$. Then, we want to solve the problem

$$\min_b b^\top \Sigma b \quad \text{subject to} \quad b^\top k = 1$$

which we can solve using Lagrange multiplies. The Lagrangian is

$$\mathcal{L}(b, \lambda) = \frac{1}{2}b^\top \Sigma b - \lambda b^\top k$$

and the derivative w.r.t. $b$ is $\Sigma b - \lambda k = 0$, which tells us that $b$ is in the direction of $\Sigma^{-1} k$. Then the constraint $b^\top k = 1$ forces the constant of proportionality, and we arrive at $b = \frac{k^\top \Sigma^{-1}}{k^\top \Sigma^{-1} k}$ $\square$

### D.4 STATEMENT AND PROOF FOR DERIVING RANK-1 EDITS FROM SUBSPACE PATCHES

**Lemma D.2.** *Given prompts A and B, two token positions $t_A$, $t_B$, and an MLP layer with output projection weight $W_{out} \in \mathbb{R}^{d_{model} \times d_{MLP}}$, let $u_A, u_B \in \mathbb{R}^{d_{MLP}}$ be the respective (post-nonlinearity) activations at these token positions in this layer. If $v$ is a direction in the activation space of the MLP layer, then there exists a ROME edit $W'_{out} = W_{out} + ab^\top$ such that the activation patch from $u_B$ into $u_A$ along $v$ and the edit result in equal outputs of the MLP layer at token $t_A$ when run on prompt A. Moreover, the ROME edit is given by*

$$a = \left((u_B - u_A)^\top v\right) W_{out}v \quad \text{and any } b \text{ that satisfies} \quad b^\top u_A = 1.$$

*Choosing $b = \frac{\Sigma^{-1} u_A}{u_A^\top \Sigma^{-1} u_A}$ minimizes the change to the model (in the sense of Meng et al. (2022a)) over all such rank-1 edits.*

---

[5]Note that in practice $W_{out}$ may be singular or poorly conditioned, because the layer normalization encourages features to sum to zero, which could to some extent also persist after a non-linearity. If this is the case, all our results apply with $\Sigma^+$ instead of $\Sigma^{-1}$.

*Proof.* The activation after patching from B into A along $v$ is $u'_A = u_A + ((u_B - u_A)^\top v)v$, which means that the change in the output of the MLP layer at this token will be

$$W_{out}u'_A - W_{out}u_A = ((u_B - u_A)^\top v)W_{out}v$$

The change introduced by a fact edit at this token is

$$W'_{out}u_A - W_{out}u_A = ab^\top u_A = \left(b^\top u_A\right)\left((u_B - u_A)^\top v\right)W_{out}v$$

and the two are equal because $b^\top u_A = 1$.

To find the $b$ that minimizes the change to the model, we minimize the variance of $b^\top x$ when $x \sim \mathcal{N}(0, \Sigma)$ subject to $b^\top u_A = 1$. The variance is equal to $b^\top \Sigma b$, so we have a constrained (convex) minimization problem

$$\min \frac{1}{2}b^\top \Sigma b \quad \text{subject to} \quad b^\top u_A = 1$$

The rest of the proof is the same as in Lemma D.1. Namely, we can solve this optimization problem using Lagrange multiplies. The Lagrangian is

$$\mathcal{L}(b, \lambda) = \frac{1}{2}b^\top \Sigma b - \lambda b^\top u_A$$

and the derivative w.r.t. $b$ is $\Sigma b - \lambda u_A = 0$, which tells us that $b$ is in the direction of $\Sigma^{-1}u_A$. Then the constraint $b^\top u_A = 1$ forces the constant of proportionality. □

In this appendix, we collect some useful notes on the rank-1 model editing (ROME) method from Meng et al. (2022a).

**Lemma D.3** (Alternative characterization of rank-1 model editing via variance minimization)**.**

### D.5 Additional experiments comparing fact patching and rank-1 editing

In Figure 14, we plot the distributions of the logit difference between the correct object for a fact and the object we are trying to substitute when patching the 1-dimensional subspaces found by DAS, and performing the equivalent rank-1 weight edit according to Lemma D.2. We observe that the two metrics quite closely track each other, indicating that the additional effects of using a weight edit (as opposed to only intervening at a single token) are negligible.

Similarly, in Figure 15, we show the success rate of the the two methods in terms of making the model output the object of the fact we are patching from. Again, we observe that they quite closely track each other.

## E   Why Do We Expect the Illusion to be Prevalent in Practice?

### E.1   MLP weights are full-rank matrices

In figure 16, we plot the 100 smallest singular values of the MLP weights in GPT2-Small for all 12 layers. We observe that they the vast majority are bounded well away from $0$. This confirms that both MLP weights are full-rank transformations.

### E.2   Features in the residual stream propagate to hidden MLP activations

**Intuition**. Suppose we have two classes of examples that are linearly separable in the residual stream. The transformation from the residual stream to the hidden MLP activations is a linear map followed by a nonlinearity, specifically $x \mapsto \text{gelu}(W_{in}x)$. As we observed in E.1, the $W_{in}$ matrix is full-rank, meaning that all the information linearly present in $x$ will also be so in $W_{in}x$. Even better, since $W_{in}$ maps $x$ from a $d_{model}$-dimensional space to a $d_{MLP} = 4d_{model}$-dimensional space, this should intuitively make it much easier to linearly separate the points, because in a higher-dimensional space there are many more linear separators. On the other hand, the non-linearity has an opposite effect: by compressing the space of activations, it makes it harder for points to be separable. So it is a priori unclear which intuition is decisive.

**Empirical validation**. However, it turns out that empirically this is not such a problem. To test this, we run the model GPT2-Small on random samples from its data distribution (we used OpenWebText-10k), and extract 2000 activations of an MLP-layer after the non-linearity. We train a linear regression with $\ell_2$-regularization to recover the dot product of the residual stream immediately before the MLP-layer of interest and a randomly chosen direction. We repeat this experiment with different random vectors and for each layer. We observe that all regressions are better than chance and explain a significant amount of variance on the held-out test set ($R^2 = 0.71 \pm 0.17, \text{MSE} = 0.31 \pm 0.18, p < 0.005$). Results are shown in Figure 17 (right) (every marker corresponds to one regression model using a different random direction).

The position information in the IOI task is really a binary feature, so we are also interested in whether *binary* information in general is linearly recoverable from the MLP activations. To test this, we sample activations from the model run on randomly-sampled prompts. This time however, we add or subtract a multiple of a random direction $v$ to the residual stream activation $u$, and calculate the MLP activations using this new residual stream vector $u'$:

$$u' = u + y \times z \times \|u\|_2 \times v$$

where $y \in \{-1, 1\}$ is uniformly random, $z$ is a scaling factor we manipulate, and $v$ is a randomly chosen direction of unit norm. For each classifier, we randomly sample a direction $v$ that we either add or subtract (using $y$) from the residual stream. The classifier is trained to predict $y$. We rescale v to match the average norm of a residual vector and then scale it with a small scalar $z$.

Then, a logistic classifier is trained on 1600 samples. Again, we repeat this experiment for different $v$ and $z$, and for each layer. We observe that the classifier works quite well across layers even with very small values of $z$ (still, accuracy drops for $z = 0.0001$). Results are shown in Figure 17 (right), and Table 2.

Table 2: Mean Accuracy for Different Values of $z$

| $z$ | Mean Accuracy |
|---|---|
| 0.0001 | 0.69 |
| 0.001 | 0.83 |
| 0.01 | 0.87 |
| 0.1 | 0.996 |

# F  SUPPLEMENTARY FIGURES

## F.1  ADDITIONAL FIGURES FOR SECTION 3

## F.2  ADDITIONAL FIGURES FOR SECTION 4

## G FIGURES ADDED DURING REVISION

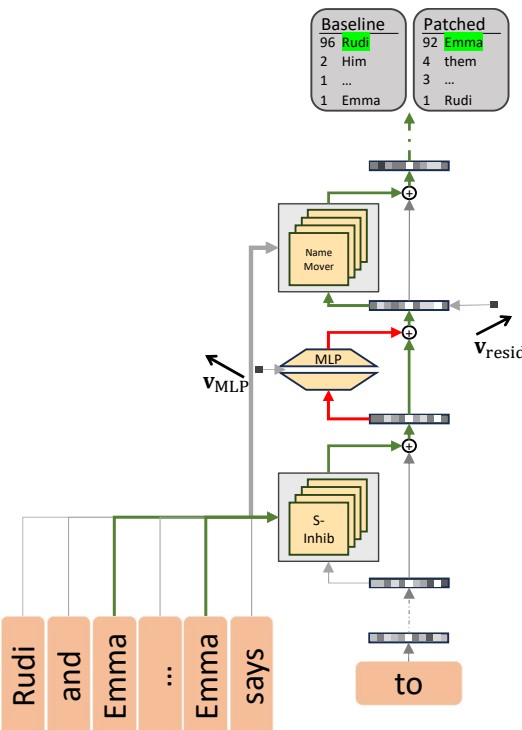

Figure 21: Schematic of the IOI circuit and interventions. GPT2-small predicts the correct name by S-inhibition heads writing positional information to the residual stream which is used by the name movers to copy the non-duplicated name (green arrows). Location of subspace interventions $\mathbf{v}_{\text{resid}}$ and $\mathbf{v}_{\text{MLP}}$ are marked. Patching the illusory subspace $\mathbf{v}_{\text{MLP}}$ adds a new path (red) along the established one that is used to flip positional information when patched.

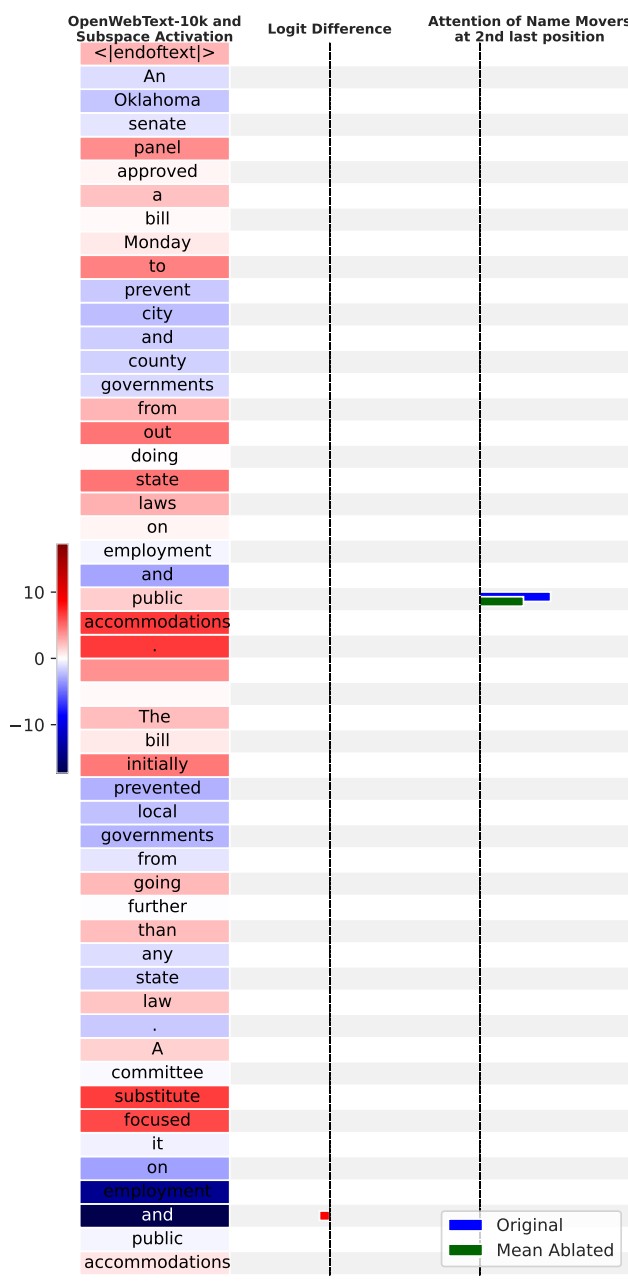

Figure 8: The IOI position subspace generalizes to arbitrary OpenWebText prompts

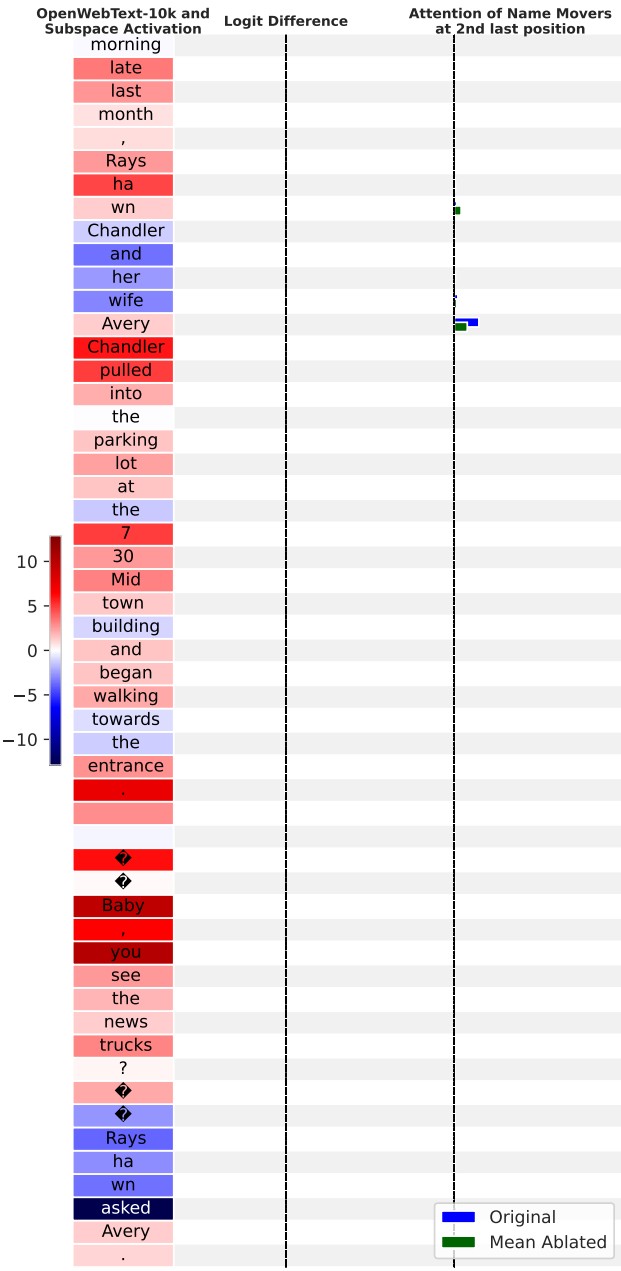

Figure 9

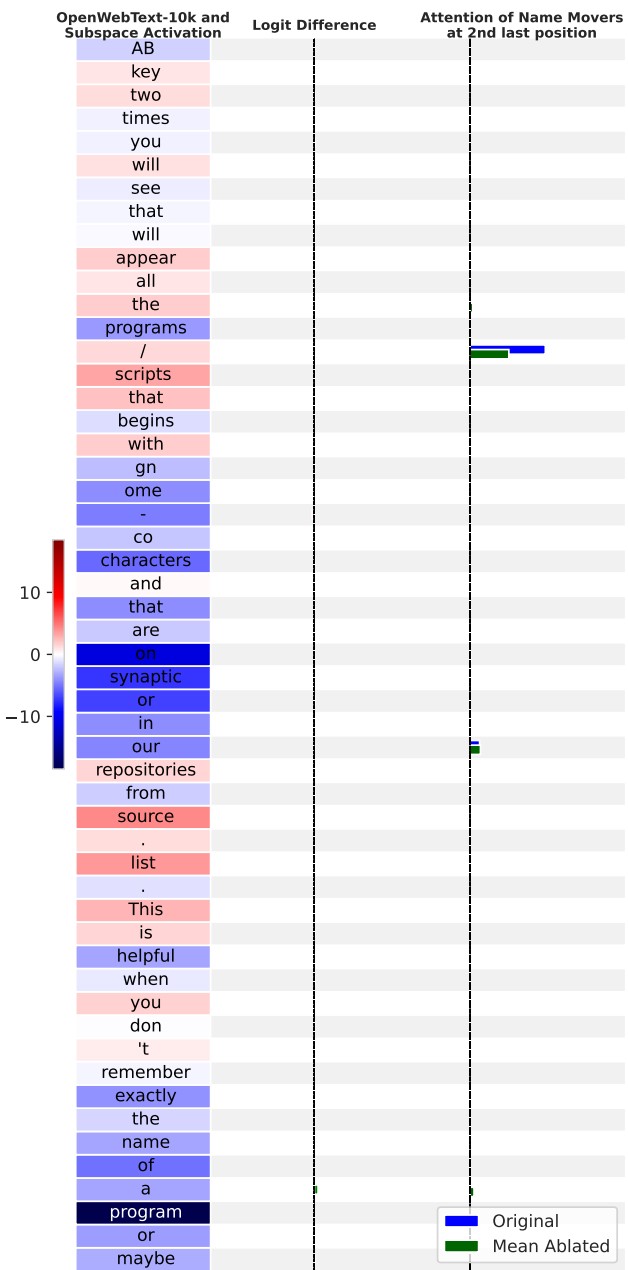

Figure 10

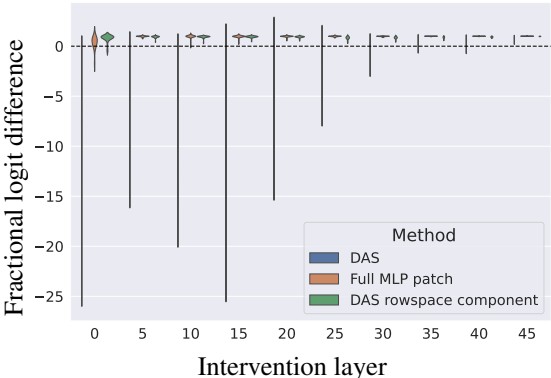

Figure 11: Fractional logit difference distributions under three interventions: patching along the direction found by DAS (blue), patching the component of the DAS direction in the rowspace of $W_{out}$ (green), and patching the entire hidden MLP activation (orange).

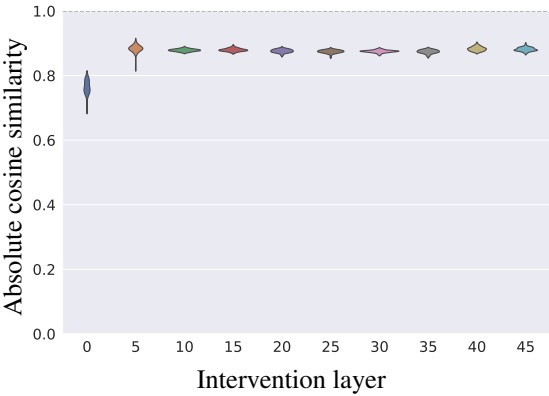

Figure 12: Distribution of the absolute value of the cosine similarity between the nullspace component of the DAS fact patching directions and the difference in activations of the last tokens of the two subjects.

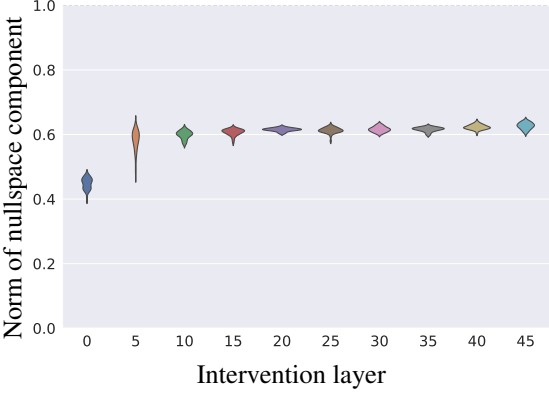

Figure 13: Distribution of the norm of the nullspace component of the DAS direction across intervention layers.

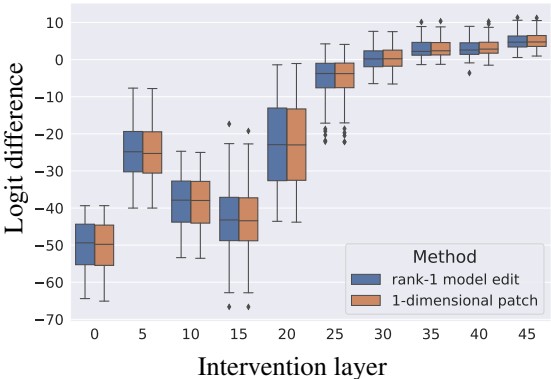

Figure 14: Comparison of logit difference between 1-dimensional fact patches and their derived rank-1 model edits

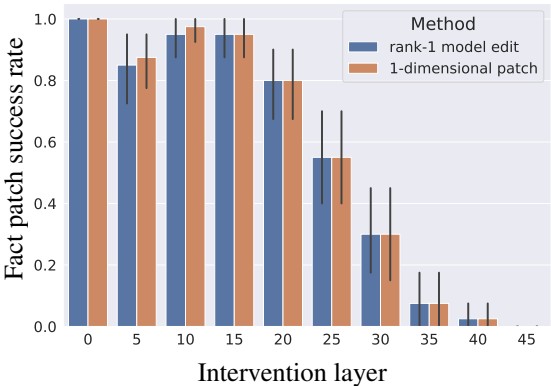

Figure 15: Comparison of fact editing success rate between 1-dimensional fact patches and their derived rank-1 model edits

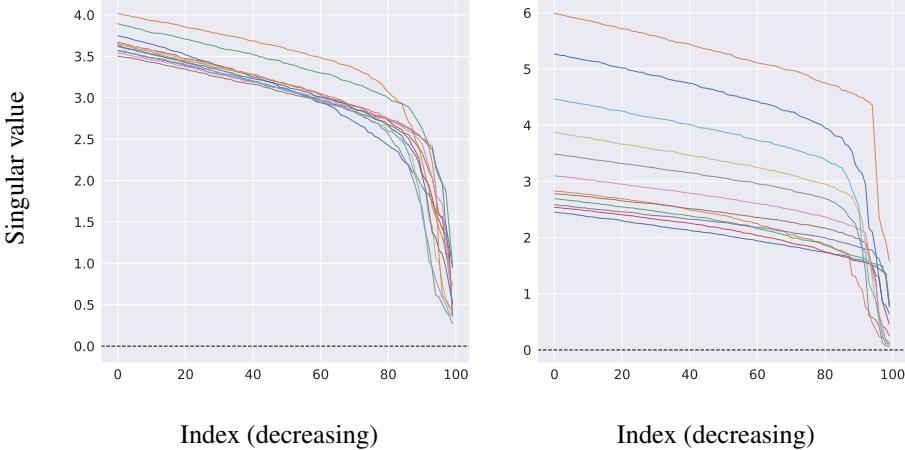

Figure 16: Smallest 100 singular values of the $W_{in}$ (left) and $W_{out}$ (right) MLP weights by layer in in GPT2-Small

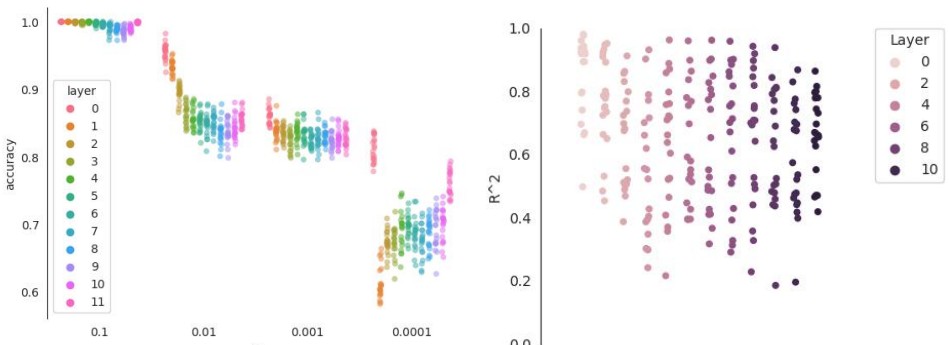

Figure 17: Recovering residual stream features linearly from hidden MLP activations: classification (left) and regression (right).

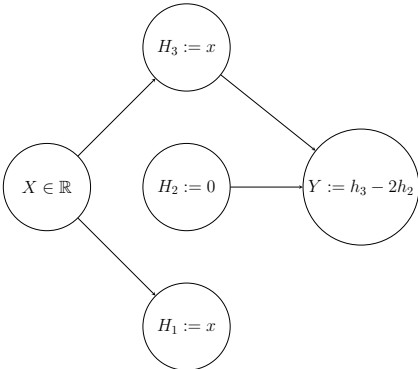

(a) The network $\mathcal{A}$ that computes the identity function. The hidden unit $H_3$ stores the value of the input and passes this to the output, while the unit $H_2$ is dormant and $H_3$ is disconnected. However, the linear subspace of $H_2$ and $H_3$ defined by the unit vector $[\frac{1}{\sqrt{2}}, -\frac{1}{\sqrt{2}}]$ fully mediates the information flow from input to output just like the unit $H_3$.

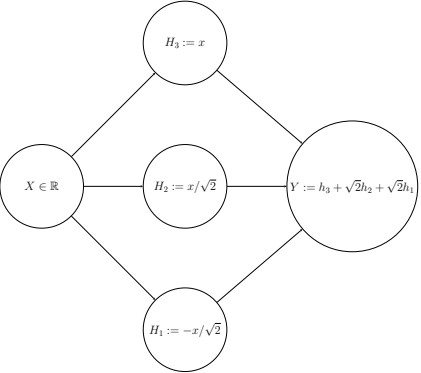

(b) The network $\mathcal{A}$ except the weights have been transformed such that the hidden units $H_2$ and $H_3$ are viewed under the coordinate vectors $[\frac{1}{\sqrt{2}}, -\frac{1}{\sqrt{2}}]$ and $[\frac{1}{\sqrt{2}}, \frac{1}{\sqrt{2}}]$. When we generalize activation patching to arbitrary subspaces, we are forced to consider this transformed network to be analytically identical to $\mathcal{A}$.

Figure 18: Diagrams of small linear networks illustrating a concrete instantiation of the interpretability illusio, alongside a subspace faithful to a model's computation.

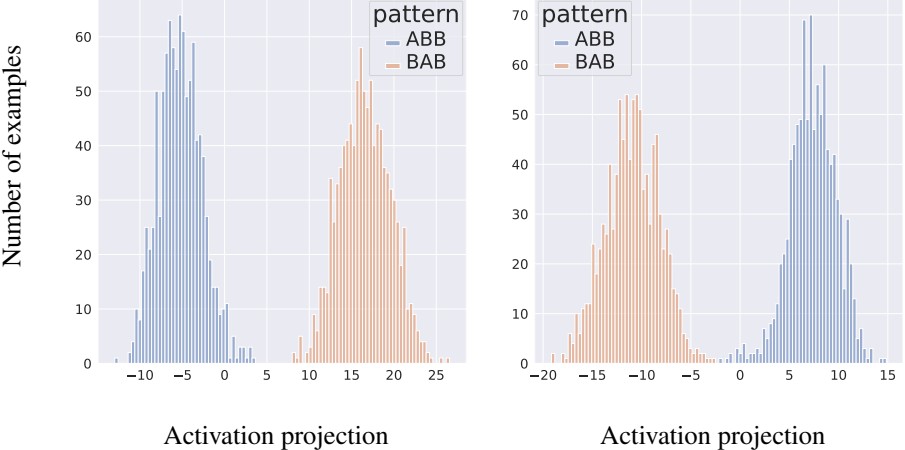

Figure 19: Projections of dataset examples' activations in the residual stream after layer 8 onto the $\mathbf{v}_{\text{resid}}$ direction found by DAS and the $\mathbf{v}_{\text{grad}}$ direction which is the gradient for difference in attention of the name mover heads to the two names.

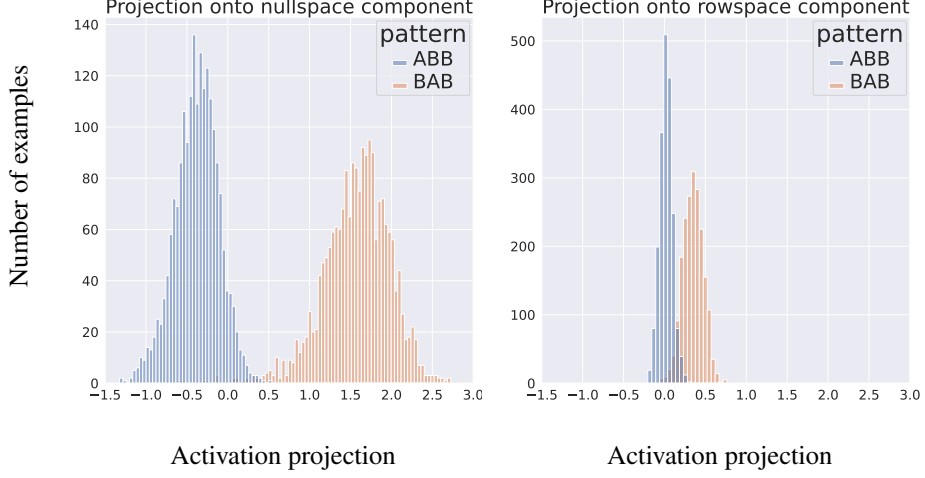

Figure 20: Projections of dataset examples onto the two components of the illusory patching direction found in MLP8: the nullspace (irrelevant) component (left), and the rowspace (dormant) component (right).

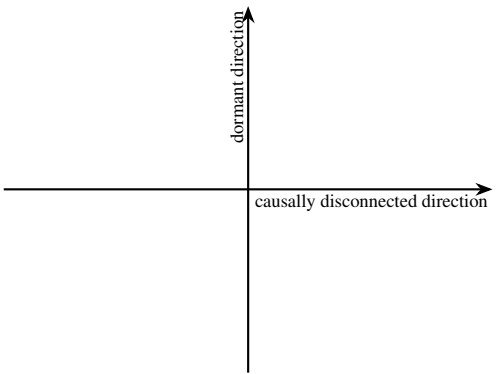

Figure 22: Consider a 2-dimensional subspace of model activations, with an orthogonal basis where the $x$-axis is *causally disconnected* (changing the activation along it makes no difference to model outputs) and values on the $y$ axis are always zero for examples in the data distribution (a special case of a *dormant* direction).

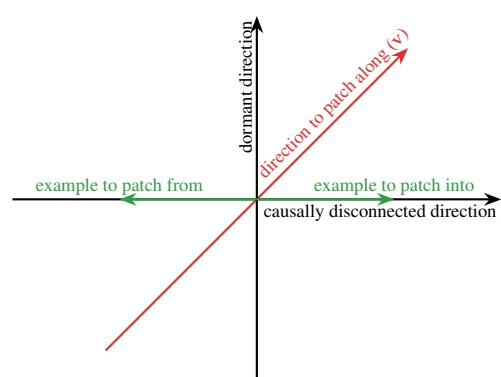

Figure 23: Suppose we have two examples (green) which differ in their projection on the causally disconnected direction (and have zero projection on the dormant direction, by definition). Let's consider what happens when we patch from the example on the left into the example on the right along the 1-dimensional subspace $v$ spanned by the vector $(1,1)$ (red)

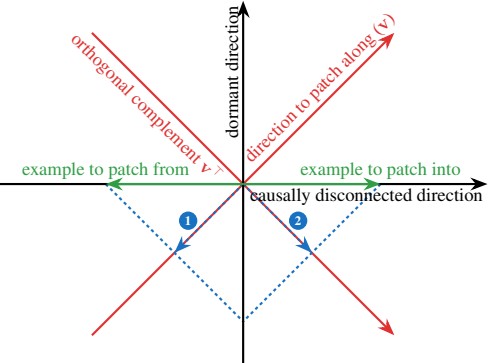

Figure 24: To patch along $v$ from the left into the right example, we match the value along $v$ from the left one, and leave the value along $v^\perp$ unchanged. In other words, we take the component of the left example along $v$ (①) and sum it with the $v^\perp$ component (②) of the original activation.

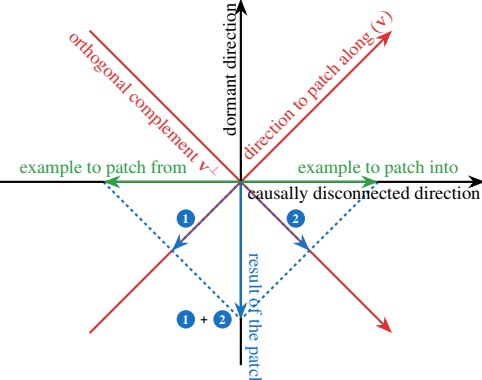

Figure 25: This results in the patched activation ① +②, which points completely along the dormant direction. In this way, the variation of activations along the causally disconnected $x$-axis results in activations along the previously dormant $y$-axis

Figure 26: A step-by-step illustration of the phenomenon shown in Figure 1.

