# OpenReview forum: "Is This the Subspace You Are Looking for? An Interpretability Illusion for Subspace Activation Patching"
_ICLR.cc/2024/Conference — ICLR 2024 poster_

### Official Review · Reviewer_ERkw · 2023-10-20

**Soundness:** 2 fair
**Presentation:** 4 excellent
**Contribution:** 4 excellent
**Rating:** 8
**Confidence:** 2

**Summary:**

This paper identifies a challenge for the approach of looking for subspaces corresponding to causal factors of deep neural network output.  It claims that activation patching and subspace patching approaches are at risk of an interpretability illusion where the patching fails to change the causal pathways actually leading to undesired input.  It develops the concepts of dormant pathways and causally disconnected pathways to explain why the interpretability illusion is possible.  The paper illustrates its model of the interpretability illusion using a toy example.  Furthermore, it presents real-life case studies.  One set of case studies focuses on language models, particularly the indirect object identification task.  For the IOI task, previous works have used activation patching to correct a network to output the indirect object of a sentence rather than the direct object.  The paper identifies an approach to the task, using distributed alignment search, that is subject to the interpretability illusion.  It also identifies an approach, using prior knowledge about the network to identify the nodes responsible for the error, that correctly patches the network.  The paper explains the difference between these approaches using the concepts of dormant and causally disconnected pathways.  Furthermore, the paper also studies factual recall in language models as an additional case study.  It presents a set of experiments where it shows that the vulnerability of activation patching to the interpretability illusion depends on the layer which is targeted.  The paper also includes some additional discussion on connections between activation patching and rank-one model edits.

**Strengths:**

This is a highly relevant work for the field of deep neural network interpretability, as it identifies a serious obstacle to the activation and subspace patching approaches.  The concepts of dormant and causally disconnected pathways are intuitive, and they are developed with the aid of theory and experiment.  The paper is clearly written, and figures help illustrate the concepts of the interpretability illusion, and the mechanisms involved in the IOI task.  This paper is significant for both identifying a practical problem and for developing intuitions that deepen our understanding of interpretability methods for deep neural networks.

**Weaknesses:**

The definitions seem rather brittle.  Requiring strict equality in the definitions of causally disconnected and dormant seems to be very limiting for practice.

**Questions:**

1. How do we know that the theoretical account for the interpretability illusion is actually the explanation for the failure of activation patching in the case studies presented?
2. Are there a set of empirical predictions (e.g. about the directions of activations or gradients) that could either falsify or support the theoretical model?
3. Why not relax the definitions of "causally disconnected" or "dormant" to not require strict equality, but rather to specify that the effect of the the patching be small within some threshold?

---

> ### Author Response · Authors · 2023-11-16
>
> We thank the reviewer for their thoughtful review, and for the encouraging
> feedback on the relevance and clarity of our work. Below, we address the
> reviewer’s questions and other concerns:
>
> > The definitions seem rather brittle. Requiring strict equality in the definitions of causally disconnected and dormant seems to be very limiting for practice. [...] Why not relax the definitions of "causally disconnected" or "dormant" to not require strict equality, but rather to specify that the effect of the the patching be small within some threshold?
>
> **We acknowledge that requiring strict equality in the definition of a dormant
> subspace is limiting. In our revision of the paper, we have changed the
> definition to allow for approximate equality instead**, which reflects the reality
> better. In our experiments (Figure 20 and Figure 12) we find that the orthogonal
> complements of the causally disconnected components of the subspaces we find are
> relatively dormant compared to the causally disconnected components.
>
> As for causally disconnected subspaces, **it is not too limiting to require strict
> equality, and all our empirical examples of causally disconnected subspaces obey
> strict equality**. Recall that a subspace of the activations of some model
> component is causally disconnected if changing the activation only along this
> subspace (leaving the component orthogonal to the subspace unchanged) does not
> change the model’s output. This assumption is provably satisfied when, for
> example, we consider the post-GELU activations of an MLP layer, and
> $v_{disconnected}$ is in the kernel of the layer’s down-projection. This is
> because the only causal pathway by which the MLP activations affect the rest of
> the model is via the layer’s down-projection. A down-projection is a linear
> operation, and adding a vector in its kernel does not change its output.
>
> > How do we know that the theoretical account for the interpretability illusion is actually the explanation for the failure of activation patching in the case studies presented? [...] Are there a set of empirical predictions (e.g. about the directions of activations or gradients) that could either falsify or support the theoretical model?
>
> We connect our theoretical model of the illusion to our experiments using the following approach:
> - Suppose we have a 1-dimensional subspace $v$ of the post-GELU activations of an
> MLP layer that we want to argue exhibits the illusion
> - Decompose $v = v_{nullspace} + v_{rowspace}$, where $v_{nullspace}$ is the orthogonal projection of $v$ on the kernel of $W_out$, and $v_{rowspace}$ is the orthogonal projection of $v$ on the orthogonal complement of the kernel of $W_out$.
> - By definition, $v_{nullspace}$ is a causally disconnected direction (as
> explained in our response to the reviewer’s concern about the definitions of
> causally disconnected and dormant subspaces). This means that the only
> downstream effect of patching along $v$ is the change of the activation along
> $v_{rowspace}$. Thus, the only consequential difference between patching along
> $v$ and patching along $v_{rowspace}$ is the magnitude of the change of the
> activation along $v_{rowspace}$.
> - Therefore, a **key experiment to check for the illusion** is to patch
> only long $v_{rowspace}$ and compare the effect to patching along $v$. If the effect
> from patching along $v$ is much stronger, this suggests the presence of the
> illusion, because the disconnected component must vary substantially in order to
> change the coefficient along $v_{rowspace}$
> - To further argue that v_rowspace is approximately dormant, we use different
> approaches for the two tasks:
>   - For the IOI task, we plot the distribution of activations, colored by the
>   feature of interest (position of the IO name in the sentence), and projected
>   on the $v_{rowspace}$ or $v_{nullspace}$ directions (Figure 20). This shows
>   that $v_{rowspace}$ is significantly more dormant than $v_{nullspace}$;
>   - For factual recall, we again decompose the directions found by DAS (which
>   are in the post-gelu activations of MLP layers) in the same way.
>     - Again by definition, the nullspace components are causally disconnected.
>     - To check how dormant the rowspace components are, we ran the experiment
>     shown in Figure 12 in the paper, where we project the difference of
>     activations of the two examples we are patching between on the causally
>     disconnected component (the orthogonal projection on $\ker W_{out}$), and
>     observe cosine similarity between this projection and the original direction
>     of ~0.9; this implies that the cosine similarity with the rowspace component
>     is ~0.4; thus, the rowspace component is relatively dormant compared to the
>     nullspace one.
>     - In our revision of the paper, we have included a note on this in the main body of the text.

---

> > ### Comment · Reviewer_ERkw · 2023-11-21
> > **Good idea for validating theory**
> >
> > Your response is helpful for thinking about experimentally testing for the validity of your theory.  You describe a test
> >
> > > A key experiment to check for the illusion is to patch only [a]long $v_{rowspace}$ and compare the effect to patching along $v$.
> >
> > Which I agree would be practical, and could be used to quantify the prevalence of the effect in various networks.  Do you think this experiment could be operationalized into a method for detecting interpretability illusions?

---

> > > ### Author Response · Authors · 2023-11-23
> > > **Response to reviewer**
> > >
> > > We believe this experiment is an important signal for the presence illusion, but
> > > we would also stress that it is somewhat specific to subspaces of hidden MLP
> > > activations.
> > >
> > > Specifically, for an MLP layer, there is a clear notion of a rowspace-nullspace
> > > decomposition of a given activation vector. Namely, since we are considering the
> > > post-GELU activations, and the only way such activations influence model output
> > > is through multiplication with the down-projection matrix $W_{out}$ of the
> > > layer, it is clear that, given an activation v, we can write $v = v_{nullspace} + v_{rowspace}$ where $v_{nullspace}\in \ker W_{out}$ and $v_{rowspace} \perp
> > > v_{nullspace}$, and we know that intervening by modifying this activation to
> > > change its projection along $v_{nullspace}$ will not change the model's outputs.
> > > This is because $W_{out}(\alpha v_{nullspace} + v_{rowspace}) =
> > > W_{out}v_{rowspace} = W_{out}(v_{nullspace}+v_{rowspace})$ for any $\alpha\in\mathbb{R}$.
> > >
> > > On the other hand, in the residual stream of a transformer, an analogous
> > > decomposition would be vacuous. This is because the residual stream is
> > > relatively low-dimensional (e.g., 768 in GPT2-Small), but at the same time it is
> > > read by many attention heads in subsequent layers. This means that it is quite
> > > likely the intersection of the nullspaces of all these attention heads'
> > > key/query matrices is an empty subspace. We have also verified this empirically.
> > >
> > > This is the reason why we consider only the query matrices of name mover heads
> > > in Section 5 of the paper. This gives us a non-vacuous notion of a
> > > nullspace-rowspace decomposition. The downside is that it relies on assumptions
> > > about the "important" parts of the model. We base these assumptions on the detailed previous work on the IOI circuit.
> > >
> > > To sum up, this experiment is best applied to subspaces of hidden activations of MLP layers.

---

### Official Review · Reviewer_8jAi · 2023-10-23

**Soundness:** 1 poor
**Presentation:** 1 poor
**Contribution:** 1 poor
**Rating:** 3
**Confidence:** 3

**Summary:**

This paper shows that subspace activation patching, a technique in mechanistic interpretability used to find subspaces that are “causally” responsible for models to produce certain outputs, may not be reliable. In particular, the paper explains why these may be misleading, involving the activation of dormant pathways. Real-world examples involving indirect object identification (IOI) and fact editing are shown where such techniques are misleading.

**Strengths:**

+ The paper presents an interesting hypothesis that subspace activation patching methods can be misleading because of dormant pathways. On an intuitive level, this is an interesting hypothesis and needs to be considered for future applications of activation patching.

**Weaknesses:**

The paper does not pose an explicit hypothesis that can be falsified, limiting its scientific validity. Further, the "illusion" seems to make **problematic assumptions** and has unclear implications.

- The hypothesis seems to be that every patched subspace “v” discovered by DAS can be decomposed into two directions “v_disconnected” and “v_dormant” (with specific properties), and that "patching along the sum of these directions, the variation in the disconnected part activates the dormant part, which then achieves the causal effect". The latter statement assumes that the model behaves somewhat linearly along these subspaces (i.e., the output of the model along direction "v" is given by a sum of its outputs along "v_disconnected" and "v_dormant"), which is a strong hypothesis given that these models are fundamentally non-linear. While the paper provides some evidence for the existence of some "disconnected" and "dormant" directions, unfortunately, I do not see evidence justifying the apparent linear behaviour of the underlying model.

- The definition of the "dormant" subspaces is confusing in the context of this work. If the model output remains unchanged for in-distribution patching and changes only for out-of-distribution samples $x,y \sim \mathcal{D}$, what procedures presented in this work result in out-of-distribution samples/activations? As far as I can tell, all procedures described involve patching with in-distribution data.

- Is the hypothesis that such a decomposition does not exist for “true” subspace directions?

---

The paper (especially sections 4-7) is difficult to read, is very dense, and has **several omitted details**.

- The main paper on its own does not seem to be self-contained and seems to contain a significant number of references to the appendix.

- The main hypothesis involves the claims that (1) subspace directions can be decomposed into two directions (disconnected, dormant) with specific properties, and (2) the effect of such directions adds somewhat linearly to the model outputs. However, the paper fails to connect this terminology ("disconnected, dormant directions") in sections 4-7, making it difficult to verify whether the experiments confirm or deny the hypothesis.

- The writing and presentation are sloppy. For example, in Section 4, “ker W_out” is never defined, and it is unclear how the results in Table 1 and Figure 3 relate to the description in section 4.3. What is ABB / BAB in Figure 3? What does “connected” in Table 1 refer to? Overall, how do Table 1, Figure 3, and Figure 4 illustrate support for the presented hypothesis?

---

Overall, while this work may present a useful point, its writing (especially sections 4-7) makes it impossible to verify this. I suspect that a thorough rewrite is necessary to clarify its central point.

**Questions:**

- For a future draft, I encourage the authors to present an explicit falsifiable hypothesis that facilitates both experimentation and analysis.

- It might be helpful to comment on the downstream applications and practical utility of such activation patching techniques and mechanistic interpretability, particularly to better understand the implications of the "illusion". What are the use cases for identifying model components (neurons/subspaces) responsible for model behavior? What evaluation metrics are available to test whether the components have been correctly identified?

- Do these results for subspace activation patching also hold for usual activation patching? It seems like subspace patching is a generalization, and thus it must, and also I see that the toy example is given for the case of usual activation patching, but the experimental results and the paper's messaging are specific to subspace patching.

---

> ### Author Response · Authors · 2023-11-16
> **Clarifying fundamental aspects of our work that were miscommunicated**
>
> We thank the reviewer for their thoughtful and thorough review. We are sorry
> that reading the paper was difficult, and we thank the reviewer for pointing out
> several confusing parts of the paper. **However, there are some fundamental
> aspects of our work that were miscommunicated, and we would like to clarify them
> below**. We have uploaded a revision that we hope will alleviate these concerns,
> with changes marked in red.
>
> > The paper does not pose an explicit hypothesis that can be falsified, limiting
> its scientific validity. [...] The hypothesis seems to be that every patched
> subspace “v” discovered by DAS can be decomposed into two directions
> “v_disconnected” and “v_dormant” (with specific properties)
>
> The key message of our work is not that the illusion described will always
> happen given some set of assumptions. Instead, **our central focus is showing
> the *existence* of the illusion in practical cases of interest** for ML
> interpretability. In the cases that we empirically discovered, the subspaces
> found indeed decompose as a sum of two orthogonal vectors with the properties
> given in Section 3 of the paper, and we found this decomposition a helpful way
> to think about the mechanistic underpinnings of the illusion. **We do not claim
> to provide specific conditions that will *guarantee* that the illusion will
> occur**, or that such a decomposition will exist. This means that our work takes
> the form of demonstrating the existence of the illusion, rather than proving a
> falsifiable hypothesis that it should always happen, which we expect is too
> strong a claim.
>
> As other reviewers have noted, this is a valuable contribution to the
> interpretability literature, as **the existence of such cases demonstrates that
> one should not apply subspace patching techniques blindly**. In particular, a
> key takeaway is that one should not rely solely on end-to-end metrics when
> evaluating optimization-based subspace patching methods for the purpose of
> localizing features in language models.
>
> We remark that we also provide several theoretical and heuristic reasons to
> expect the illusion to be prevalent in practice in Section 7, but they are not
> load-bearing to our main message.
>
> > ..."patching along the sum of these directions, the variation in the
> disconnected part activates the dormant part, which then achieves the causal
> effect". The latter statement assumes that the model behaves somewhat linearly
> along these subspaces (i.e., the output of the model along direction "v" is
> given by a sum of its outputs along "v_disconnected" and "v_dormant"), which is
> a strong hypothesis given that these models are fundamentally non-linear.
>
> **We make no assumptions of linearity, only about the behavior of the model when
> changing the activation only along v_disconnected, or only along v_dormant**. In
> particular, to address the reviewer’s objection, we only need a weaker set of
> assumptions: that **changing the activation only along v_disconnected does not
> change model outputs**. This assumption is provably satisfied when, for example,
> we consider the post-GELU activations of an MLP layer, and v_disconnected is in
> the kernel of the layer’s down-projection. This is because the only causal
> pathway by which the MLP activations affect the rest of the model is via the
> layer’s down-projection. A down-projection is a linear operation (though
> subsequent model layers are not linear), and adding a vector in its kernel does
> not change its output, thus adding v_dormant, or adding v_dormant and
> v_disconnected must have exactly the same downstream effect.
>
> To unpack this in more detail, we believe that in the review, the words “...the
> output of the model along direction "v" is given by a sum of its outputs along
> "v_disconnected" and "v_dormant"...” are intended to mean “when changing the
> activation along v, this results in the same change in model output as the sum
> of the changes when changing only along v_disconnected and when changing only
> along v_dormant, adjusting the magnitudes of the changes by the projection of v
> on the two vectors”.
>
> **This is a true mathematical fact, but it is independent of any linearity
> assumptions on the neural network**. Recall that by construction v =
> (v_disconnected + v_dormant) / sqrt(2). Changing some activation A along v alone
> is the same as changing it along v_disconnected and v_dormant simultaneously
> (with coefficient 1/sqrt(2)), and leaving it unchanged along all directions
> orthogonal to the plane spanned by v_disconnected and v_dormant. Let’s call this
> new activation A’. By assumption, changing the activation A’ along
> v_disconnected alone results in no change in model output; so this activation A’
> will produce the same model output as the activation A’’ where only v_dormant
> was changed. This shows that the change in outputs introduced by changing along
> v is the same as the change in outputs introduced by changing along v_dormant
> alone, plus the change in outputs after changing along v_disconnected alone
> (which is zero).

---

> ### Author Response · Authors · 2023-11-16
> **Response on dormant/disconnected directions, and connecting theory to experiments**
>
> Continuing from our previous response, we would like to clarify some other aspects of our work that were miscommunicated.
>
> > The definition of the "dormant" subspaces is confusing in the context of this work. If the model output remains unchanged for in-distribution patching and changes only for out-of-distribution samples, what procedures presented in this work result in out-of-distribution samples/activations? As far as I can tell, all procedures described involve patching with in-distribution data.
>
> While the reviewer’s concern would be true in the case of full component activation patching, **it is possible for subspace activation patching between in-distribution data to result in out-of-distribution activations** when the subspace is a proper subspace of activation space.
>
> To explain in more detail, imagine a 2-dimensional activation space, where
> in-distribution activations are always on the x-axis, ie the y-coordinate is
> always zero. Consider subspace activation patching along the vector
> $v=(1/\sqrt(2), 1/\sqrt(2))$ which makes equal angles with the x and y axes. If
> we patch along $v$ from an example with activation $(x_1, 0)$ into an example
> with activation $(x_2, 0)$ with $x_1 \neq x_2$, the result is the activation
>
> $$(x_2, 0) + 1/\sqrt(2) * (x_1-x_2) * v =  ((x_1+x_2)/2, (x_1-x_2)/2)$$
>
> In particular, this activation is now out-of-distribution, since it has a
> nonzero component along the y axis.
>
> > Is the hypothesis that such a decomposition does not exist for “true” subspace directions?
>
> **A definition of “true” subspace directions is outside the scope of our work**.
> Rather, our central message is that the existence of such a decomposition, and
> an empirical observation showing that removing the causally disconnected
> component of a subspace greatly reduces the effect of the subspace activation
> patch, should be taken as cause for concern when attributing features to
> particular subspaces.
>
> > [...] However, the paper fails to connect this terminology ("disconnected, dormant directions") in sections 4-7, making it difficult to verify whether the experiments confirm or deny the hypothesis.
>
> We do connect the terminology of disconnected and dormant directions to our experiments in several ways:
> - For the IOI task, we orthogonally decompose the $v_{MLP}$ subspace as a sum $v_{MLP} = v_{MLP}^{nullspace} + v_{MLP}^{rowspace}$, where $v_{MLP}^{nullspace}$ is the orthogonal projection of $v_{MLP}$ on the kernel of $W_{out}$, and $v_{MLP}^{rowspace}$ is the orthogonal projection on the rowspace of $W_{out}$.
>   - By definition, $v_{MLP}^{nullspace}$ is a causally disconnected direction.
>   - We show that the direction $v_{MLP}^{rowspace}$ is significantly less activated by the feature we are patching than the direction $v_{MLP}^{nullspace}$ in Figure 20 in the Appendix. This shows that $v_{MLP}^{rowspace}$ is relatively dormant, compared to $v_{MLP}^{nullspace}$.
>   - In our revision of the paper, we have rewritten large parts of Subsection 4.3. to clarify the relationship of the experiments to our model of the illusion.
> - For factual recall, we again decompose the directions found by DAS (which are in the post-gelu activations of MLP layers) in the same way.
>   - Again by definition, the nullspace components are causally disconnected.
>   - To check how dormant the rowspace components are, we ran the experiment
>   shown in Figure 12 in the paper, where we project the difference of
>   activations of the two examples we are patching between on the causally
>   disconnected component (the orthogonal projection on $\ker W_{out}$), and
>   observe cosine similarity between this projection and the original direction
>   of ~0.9; this implies that the cosine similarity with the rowspace component
>   is ~0.4; thus, the rowspace component is relatively dormant compared to the
>   nullspace one.
>   - In our revision of the paper, we have included a note on this in the main body of the text.

---

> ### Author Response · Authors · 2023-11-16
> **Response on clarity of writing and self-containedness**
>
> > The main paper on its own does not seem to be self-contained and seems to contain a significant number of references to the appendix.
>
> We acknowledge that there is quite a bit of material in the appendix, and are
> sorry that this made our work harder to follow. We have tried to condense the
> central points of our work in the main body of the paper by making the following
> changes, which have significantly reduced the dependence of the main body in
> sections 4 and 6 on the supplementary material by including the following
> details in the main body:
> - IOI dataset we used (Section 4.1)
> - Computing the direction $v_{grad}$ (Section 4.2)
> - Computing directions using DAS (Section 4.2)
> - Details for fact patching experiments (Section 6.1)
> - We moved what was previously Figure 2 to the supplementary material, as we felt that it makes an insufficient contribution relative to the space it occupies.
> - Similarly, we moved the statement of what was previously Lemma 6.1. To the
>   appendix, and explained its result informally
>
> We hope this alleviates your concern. If there are any other parts of the
> appendices which felt important to understanding the main text, we would
> appreciate you bringing them to our attention.
>
> > The writing and presentation are sloppy. For example, in Section 4, “ker W_out” is never defined, and it is unclear how the results in Table 1 and Figure 3 relate to the description in section 4.3. What is ABB / BAB in Figure 3? What does “connected” in Table 1 refer to? Overall, how do Table 1, Figure 3, and Figure 4 illustrate support for the presented hypothesis?
>
> We have addressed these issues as well as others pointed out by the other reviewers in our updated draft. In particular, we rewrote Subsection 4.3 to fix various problems and better connect the writing to the table and figure. To answer the reviewer’s questions specifically:
> - $\ker W_{out}$ is the kernel of the down-projection of the MLP layer where the
>   illusory direction v_MLP is contained.
> - The main text now contains detailed descriptions of all interventions involved, with short names matching those appearing in Table 1.
> - Figure 3 relates to the text in Section 4.3 through the paragraph now titled
> “Patching $v_{\text{MLP}}$ activates a dormant pathway through the
> MLP.” Specifically, Figure 3 illustrates how the subspace activation patch along
> $v_{MLP}$ takes the output of the MLP layer along the gradient direction of the
> name mover query matrices off-distribution.
> - Here and elsewhere in the paper, “ABB” means prompts where the IO name comes
> first, and “BAB” means prompts where the S name comes first. We thank the
> reviewer for pointing out this omission. The notation is now explained in the
> caption of Figure 3.
>
> Overall, Table 1 supports our claim for the existence illusion by showing that
> the subspace patch along $v_{MLP}$ has a significant effect on model outputs
> (much stronger than the effect of patching the full MLP layer’s activations),
> but this effect disappears when we restrict to the causally relevant component
> of $v_{MLP}$, denoted $v_{MLP}^{rowspace}$ in our revision and introduced in the
> “Methodology” part of Subsection 4.3.  Figure 3 further supports our model of
> the illusion by showing how the MLP layer’s output on the causally relevant
> direction $v_{grad}$ (given by the gradient of name-mover attention scores) is
> taken significantly out of distribution by the patch along $v_{MLP}$. Finally,
> Figure 4 (which is about factual recall) has similar message as Table 1: it
> shows that subspace patching along the directions found by DAS is much more
> effective at changing a model’s completion of a fact than patching the entire
> MLP layer, or patching along only the causally relevant component.

---

> ### Author Response · Authors · 2023-11-16
> **Response on motivation for mechanistic interpretability and other related work**
>
> > It might be helpful to comment on the downstream applications and practical utility of such activation patching techniques and mechanistic interpretability, particularly to better understand the implications of the "illusion". What are the use cases for identifying model components (neurons/subspaces) responsible for model behavior?
>
> Mechanistic interpretability has been used in several downstream applications, some of which are: removing toxic behaviors from a model while otherwise preserving performance by minimally editing model weights (Li et al [1]), changing factual knowledge encoded by models in specific components to e.g. enable more efficient fine-tuning in a changing world (Meng et al [2]), improving the truthfulness of LLMs at inference time via efficient, localized inference-time interventions in specific subspaces (Li et al [3]), and studying the mechanics of gender bias in language models (Vig et al [4]).  **We have added these references to the introduction of the paper in our revision.**
>
> [1] Maximilian Li, Xander Davies, and Max Nadeau. Circuit breaking: Removing model behaviors with targeted ablation. In DeployableGenerativeAI, 2023
>
> [2] Kevin Meng, David Bau, Alex J Andonian, and Yonatan Belinkov. “Locating and
> editing factual associations in GPT”. In: Advances in Neural Information Processing
> Systems. 2022.
>
> [3] Kenneth Li, Oam Patel, Fernanda Viégas, Hanspeter Pfister, and Martin Wattenberg. 2023b. Inference-time intervention: Eliciting truthful answers from a language model. arXiv preprint arXiv:2306.03341.
>
> [4] Jesse Vig, Sebastian Gehrmann, Yonatan Belinkov, Sharon Qian, Daniel Nevo, Simas
> Sakenis, Jason Huang, Yaron Singer, and Stuart Shieber. “Causal mediation analysis
> for interpreting neural nlp: The case of gender bias”. In: arXiv preprint arXiv:2004.12265
> (2020).
>
> > What evaluation metrics are available to test whether the components have been correctly identified?
>
> **This is an area of active research to which our work contributes, though our work is far from the final word on the matter**. Some concrete and general techniques to evaluate LLM interpretations have been proposed, such as causal scrubbing (Chan et al. [5], mentioned in the related work of our paper). Other common techniques are based on ablations, such as (full component) activation patching, as discussed in our work. More recently, techniques like DAS have been proposed that use subspace activation patching to evaluate mechanistic interpretations of LLMs. Part of our contribution is exhibiting a situation where full-component and subspace activation patching reach different conclusions about the mechanics of the model’s behavior, and arguing why the subspace-level explanation is misleading.
>
> [5] Lawrence Chan, Adrià Garriga-Alonso, Nicholas Goldowsky-Dill, Ryan Greenblatt, Jenny Nitishinskaya, Ansh Radhakrishnan, Buck Shlegeris, and Nate Thomas. “Causal Scrubbing: a method for rigorously testing interpretability hypotheses”. https://www.alignmentforum.org/posts/JvZhhzycHu2Yd57RN/causal-scrubbing-a-method-for-rigorously-testing

---

> ### Author Response · Authors · 2023-11-16
> **Response on full-component vs subspace activation patching**
>
> > Do these results for subspace activation patching also hold for usual activation patching? It seems like subspace patching is a generalization, and thus it must, and also I see that the toy example is given for the case of usual activation patching, but the experimental results and the paper's messaging are specific to subspace patching.
>
> The toy example is of a linear neural network with one hidden layer, where we intend the layer to be a single “component” (the same way we consider an MLP’s post-GELU activations at a given token and layer in an LLM to be a single “component”). This is admittedly a matter of semantics in this simple example. We have added a clarification about this in our revision.
>
> (The following two paragraphs also appear in the response to reviewer **YsUK**) This is a great question. **A key property of subspace activation patching not shared with full-component activation patching is that it may create out-of-distribution activations for the given component, even when patching only between in-distribution examples**. This makes it particularly vulnerable to an MLP-in-the-middle type of illusion, where we take the output of the MLP layer off-distribution in order to write some causally important information in the residual stream. Intuitively, this would not be possible to such an extent when patching the full hidden activation of the MLP layer, because its activations for in-distribution examples do not generally write important information to the residual stream (as can be confirmed by doing the full patch).
>
> However, full-component activation patching can still take the activations of the model as a whole off-distribution. It is an interesting question for future work whether a variant of the illusion can be exhibited for full-component patching. For example, if two heads always cancel each other out (one outputs +v and the other -v), we could patch just one of them, such that they no longer cancel out. Thus, it may be possible to patch only one of these components and throw the model off-distribution. However, we expect such scenarios of “perfect cancellation” to be rarer in practice.

---

> ### Author Response · Authors · 2023-11-16
> **Concluding response**
>
> We apologize again for the miscommunication and lack of clarity in our writing. We have tried to clarify the parts of the paper flagged by the reviewer in our updated revision. We hope that these changes and our detailed responses alleviate the reviewer’s concerns.

---

> ### Author Response · Authors · 2023-11-21
> **Reminder about end of discussion period**
>
> Dear reviewer **8jAi**, this is a gentle reminder that the end of the discussion
> period draws near (in approximately 48 hours). **We have responded with a
> rebuttal to your comments**, and we hope you will respond back, letting us know
> if your concerns have been alleviated. If you have any remaining concerns, we
> would be happy to continue the discussion.

---

> ### Comment · Reviewer_8jAi · 2023-11-21
>
> I thank the authors for their detailed response. While I carefully read the updated paper and rebuttal, I want to clarify the following aspects:
>
> **Re: falsifiable hypothesis**: Note that a falsifiable hypothesis does not require specifying the precise conditions under which the illusion occurs. In this case, a hypothesis can also be stated as "there exist pre-trained ML models such that the following property holds with their activations: ...". I believe that presenting a falsifiable hypothesis falls well within the scope of this work and, in general, any scientific work. The key idea is to present a concrete test to validate the hypothesis (i.e., the provided explanation) BEFORE looking at the experimental evidence.
>
> **Re: definition of dormant directions**: The paper defines a dormant direction as "We say U is dormant if $M_{U_c ←u_y} (x) ≈ M(x)$ with high probability over x, y ∼ D, but not over any x, y". This definition critically relies upon having access to in-distribution and out-of-distribution inputs "x". Yet the rebuttal states that "While the reviewer’s concern would be true in the case of full component activation patching, it is possible for subspace activation patching between in-distribution data to result in out-of-distribution activations when the subspace is a proper subspace of activation space".
>
> (1) This is a contradiction: does the definition apply to inputs or patched activations, or likely, both? (If so, please clarify in the main text)
>
> (2) The presented arguments are circular. The authors state that patching creates OOD activations, yet the definition is about model output invariance in the presence of patching. Can you please clarify?
>
> (3) Why doesn't full activation patching, similar to subspace patching, also result in OOD activations? Both these techniques create "synthetic" activations, and thus, there is a high chance of both being OOD.

---

> > ### Author Response · Authors · 2023-11-22
> > **Re: definition of dormant directions**
> >
> > We thank the reviewer for their response.
> >
> > **Re: definition of dormant directions**: We apologize for the confusion and thank the reviewer for pointing it out. The definition should be: "We say [the subspace of the activation space of a model component] U is dormant if  $M_{U_{C} \leftarrow u_y}(x)\approx M(x)$ with high probability over $x, y \sim D$, but, given $x$ from $D$, there exists some $u$ such that $M_{U_{C} \leftarrow u}(x)\not \approx M(x)$”
> >
> > That is,
> > - running the model on an input $x$ from $D$, and intervening on this run by setting the projection of the activation on $U$ to that of another input from $D$ does not change model outputs much compared to just running the model on $x$ without an intervention,
> > - however, it is possible to run the model on an input $x$ from $D$, and intervene by setting the activation to some other value $u$ (which we referred to as an out-of-distribution value), and get a very different result than just running the model on $x$. Crucially, this other value $u$ cannot be obtained by running the model on some input $x$ from $D$, and projecting the activation of the subspace $U$.
> >
> > We have published a revision of the paper correcting this; the correct definition is the one we intended in all experiments and other discussions; this is an isolated bug in the paper for which we apologize.
> >
> > To address the reviewer’s concerns specifically:
> >
> > (1) **There is no contradiction**; the definition and the comment in the rebuttal refer to two related, but crucially different phenomena.
> > - The definition of a dormant subspace is as stated above. It considers two cases: one case is about subspace activation patching between in-distribution inputs; the other case is intervening on the run of the model on an in-distribution input by setting the projection of the activation of the given component to a given value $u$, which has the property that there is no input $x'$ from $D$ such that the projection of the given component's activation on $U$ when the model is run on $x'$ is $u$.
> > - The comment in the rebuttal is correct as stated. However, this comment is **not about patching a dormant subspace**. Instead, the comment refers to the key observation that there exist subspaces $S$ of the activation space of a component $C$ which, when activation-patched, can lead to an activation $A$ such that $A$'s projection on **a dormant subspace $U$ which is different from $S$** is off-distribution, i.e. there is no input $x'$ from $D$ such that the projection of the given component's activation on $U$ when the model is run on $x'$ is $u$. This is illustrated in Figure 1 in the paper, where activations over $D$ vary only along the x-axis of a two-dimensional activation space, we patch along the direction y=x, and the resulting activation is completely along the y-axis (which cannot be realized by simply inputting examples from $D$).
> >
> > (2) **Our arguments are not circular**: A dormant direction in the activation space of a model component is defined as one that can potentially change model output significantly if set to some value. In this sense, a dormant direction can **potentially** lead to non-invariant model outputs. Furthermore, as explained in (1), **the key phenomenon behind the illusion relies on patching along a subspace that is not purely dormant**, but the patch has the effect of taking the projection of the activation on the dormant subspace off-distribution.
> >
> > (3) When we say that full component activation patching does not result in OOD activations, we mean **restricting to activations of the given component only**. In this sense, full-component activation patching always exchanges one input’s activation with another’s, so the result is still in-distribution. By contrast, subspace patching can create an activation for this component which is not realizable when we use as input any example in the distribution. However, if we consider the collective activations of all components, it is true that full-component activation patching can take this collective activation off-distribution. We have also addressed this point at more length in our “Response on full-component vs subspace activation patching”.

---

> > ### Author Response · Authors · 2023-11-22
> > **Re: falsifiable hypothesis**
> >
> > We thank the reviewer for their clarification. In light of this clarification, our hypothesis can be expressed as: “There exist pre-trained transformer language models, pairs of distributions $D_{base}$ and $D_{source}$ over inputs to these language models, ground-truth next-word predictions for inputs in $D_{base} \cup D_{source}$, and 1-dimensional subspaces $S$ of post-GELU activations of MLP layers in these models, such that activation patching from $x_{source}\sim D_{source}$ into $x_{base}\sim D_{base}$ along $S$ has a strong effect of shifting probability from the ground-truth completion of $x_{base}$ to that of $x_{source}$, but this effect is significantly diminished when activation patching is performed along the component of $S$ orthogonal to the nullspace of the down-projection $W_{out}$ of the MLP layer.” Here, the nullspace component of $S$ is the causally disconnected subspace, and the rowspace component (the one orthogonal to the nullspace component) is hypothesized to possibly be dormant.
> >
> > We have included it in Section 3 of the paper in the revision recently uploaded, and our experiments in sections 4,5,6 refer directly to this (when patching along the "rowspace" component of the subspaces considered).

---

### Official Review · Reviewer_YsUK · 2023-10-27

**Soundness:** 4 excellent
**Presentation:** 4 excellent
**Contribution:** 4 excellent
**Rating:** 8
**Confidence:** 3

**Summary:**

This work finds and shows that subspace activation patching may be subject to interpretability illusions caused by activating a dormant pathway (i.e., does not respond to input changes) but is activated by a causally disconnected feature (i.e., is activated by the input change but is not connected to the model’s output). They show that this illusion also relates to rank-one fact editing (e.g., Meng et al. [1]) and explains its recently found inconsistencies [2]. Finally, they demonstrate that further analysis, e.g., manual circuit analysis, can mitigate the interpretability illusion.

[1] Meng, Kevin, et al. "Locating and editing factual associations in GPT." NeurIPS 2022.

[2] Hase, Peter, et al. "Does localization inform editing? surprising differences in causality-based localization vs. knowledge editing in language models." arXiv 2023.

**Strengths:**

* Given the rising popularity of mechanistic interpretability and (subspace) activation patching, this paper addresses a very important and significant issue in a timely manner.

* The interpretability illusion is well-motivated and clearly introduced. The formal definition (p. 5) is sound. It is also clear that the illusion is not a mere artifact of the chosen experimental settings but is present in most cases with high probability.

* The experimental design to showcase the interpretability illusion is well-designed.

* The discussion on how one can prevent the interpretability fallacy is important and sound. It paves a way for future work that relies on automatic activation patching methods, to avoid false interpretations of model behavior.

* The discussion on the presence of the interpretability illusion from a mechanistic viewpoint is sound.

* The paper is clearly written, (mostly) easy to follow, and self-containing.

**Weaknesses:**

This paper is a very strong submission. It is well-motivated, sound, and clear. The only “major” weakness is that code is not provided and minor comments (see below).

**Questions:**

* Why do the authors solely focus on subspace activation patching? The identified interpretability illusion should also hold for (automatic) component activation patching (i.e., consider the entire activation space as the (sub)space).

* There seems to be an error in the indices in the toy example in Appendix A.3.

## Suggestions

* While Fig. 1 clearly demonstrates the interpretability illusion, it is hard to parse. It may be good to make it more accessible/easier to parse, as it demonstrates the main insight of the paper.

* It’d be good to add the relation of the vector $v$ to the subspace $U$ in Sec. 3.

* It’d be good to match the notation of Tab. 1 and the respective text.

---

> ### Author Response · Authors · 2023-11-16
>
> We thank the reviewer for their thoughtful review, and for the encouraging
> feedback on the timeliness and soundness of our work. Below, we address the
> reviewer’s questions and other concerns:
>
> > code is not provided
>
> We are working on releasing an anonymized version of our code in the next few
> days. We will provide a link to it in a subsequent response once it is
> available.
>
> > Why do the authors solely focus on subspace activation patching? The
> identified interpretability illusion should also hold for (automatic) component
> activation patching (i.e., consider the entire activation space as the
> (sub)space).
>
> This is a great question. A key property of subspace activation patching not
> shared with full-component activation patching is that it **may create
> out-of-distribution activations for the given component, even when patching only
> between in-distribution examples**. This makes it particularly vulnerable to an
> MLP-in-the-middle type of illusion, where we take the output of the MLP layer
> off-distribution in order to write some causally important information in the
> residual stream. Intuitively, this would not be possible to such an extent when
> patching the full hidden activation of the MLP layer, because its activations
> for in-distribution examples do not generally write important information to the
> residual stream (as can be confirmed by doing the full patch).
>
> However, full-component activation patching can still take the activations of
> the model as a whole off-distribution. It is an interesting question for future
> work whether a variant of the illusion can be exhibited for full-component
> patching. For example, if two heads always cancel each other out (one outputs +v
> and the other -v), we could patch just one of them such that they no longer
> cancel out. Thus, it may be possible to patch only one of these components and
> throw the model off-distribution. However, we expect such scenarios of “perfect
> cancellation” to be rarer in practice.
>
> > There seems to be an error in the indices in the toy example in Appendix A.3.
>
> This is correct; the sentence “... the linear subspace of H_2 and H_3 defined by
> the unit vector…” should be “...H_1 and H_2…” instead. We thank the reviewer for
> pointing this out, and we have corrected it in our revision of the paper.
>
> > While Fig. 1 clearly demonstrates the interpretability illusion, it is hard to
> parse. It may be good to make it more accessible/easier to parse, as it
> demonstrates the main insight of the paper.
>
> We acknowledge that Figure 1 is somewhat involved and thank the reviewer for
> pointing this out. To make it more readable, we have included a “decomposition”
> of this figure into four figures to be read sequentially. Due to space
> limitations, we include this new figure as Figure 26 in the Appendix, and we
> point to it from the main text. We also re-did the original figure in TikZ to
> improve readability. We hope that this makes the phenomenon clearer.
>
> > It’d be good to add the relation of the vector v to the subspace U in Sec. 3.
>
> We thank the reviewer for pointing this out; the relationship is that, in the
> special case of 1-dimensional subspace patching we consider, $U$ is the subspace
> spanned by the (unit) vector $v$. We hope that clarifies this part of the paper;
> we have added it to our revision.
>
> > It’d be good to match the notation of Tab. 1 and the respective text.
>
> We thank the reviewer for pointing this out. This is a concern shared by another
> reviewer as well, and we acknowledge that Table 1 can be improved in many ways.
> In the revision we have uploaded, we have made the notation more coherent. We
> have also re-parametrized the fractional logit diff metric to make it more
> intuitive and readable.

---

> > ### Comment · Reviewer_YsUK · 2023-11-21
> > **Re: Official Comment by Authors**
> >
> > I thank the authors for the thorough reply. I appreciate the effort of the authors to provide code and strongly encourage them to do so. I would also encourage the authors to include Fig. 26 instead of Fig. 1 in the main text but understand that page limit may be hindering in doing so. Lastly, I appreciate the discussion on subspace vs. entire activation space patching that could be also featured in the paper.
> >
> > Overall, I believe that this paper constitutes a very important contribution to the interpretability community and consequently should be highlighted as such at the conference.

---

> > > ### Author Response · Authors · 2023-11-22
> > > **Response to reviewer**
> > >
> > > We thank the reviewer again for their encouraging feedback. We have uploaded a zip file of code for the main experiments in the paper with notebooks to reproduce key figures as a revision. We will continue working to eventually publish the code on github.

---

### Author Response · Authors · 2023-11-16
**Overall response**

We thank all three reviewers for their thoughtful feedback and insightful comments.

As reviewers **YsUK** and **ERkw** have noted, our core contribution is showing the existence of a particular interpretability illusion for subspace activation patching in two real-world examples, where a subspace seems causally relevant by activating a previously dormant pathway. The impact of this is that subspace activation patching interpretability techniques, especially when based on optimization, should not be applied using solely end-to-end evaluations, and may be misleading.

As such, reviewer **8jAi**’s concern about the lack of a “falsifiable hypothesis” that would guarantee the existence of the illusion under some conditions, is not the lens through which we view our contribution. We have addressed this, as well as other key aspects of our work that were miscommunicated, in our response to reviewer **8jAi**.

A concern shared by the majority of reviewers is about the clarity of our paper, especially sections 4-7, and about how the experiments therein connect to our theoretical model of the illusion. We have taken these comments to heart, and have updated the writing, tables and figures in these sections with clearer explanations and additional details (which were previously in the supplementary material) to aid the grounding of our experiments.

We have uploaded a revision of our paper where new material is highlighted in red to aid reviewers. Specific changes made are listed in individual responses to reviewers that requested these changes. The main changes have been:
- adding related work motivating downstream uses of mechanistic interpretability in the introduction;
- re-doing Figure 1 for improved readability, as well as providing a step-by-step decomposition and explanation for it in the Appendix
- rewriting Section 4.3. significantly to clarify experimental methodology and how it connects with the figures and tables;
- decreasing the main body's dependence on the supplementary material in Sections 4 and 6.

---

> ### Author Response · Authors · 2023-11-23
> **Update to overall response**
>
> As the discussion period nears its end, we would like to thank reviewers for their participation in the discussion. After a clarifying discussion with reviewer **8jAi**, we have included a hypothesis about the existence of the illusion in the main body of the paper (Section 3).
>
> There are still some aspects of our work that were miscommunicated in the discussion with reviewer **8jAi** after our initial response. We have posted new responses that we hope clarify these aspects.
>
> In conclusion, we believe we have identified and addresses the limitations of our work pointed out by the three reviewers, and we thank them again for the productive discussion.

---

### Meta-Review · Area_Chair_nbig · 2023-12-14

**Metareview:**

Localizing where some features or facts are stored in a language model is an important topic in the LLM era. One method for localizing facts is Activation Patching (AP), which measures the causal relationship between some unit in the LLM with its behavior e.g. factual recall, question-answering result, or gender bias. AP replaces the feature corresponding to some token in the input sequence of LLMs with the feature of a different, real token to measure the importance of a given token to some model outputs or behaviors.

The paper's main result is an explanation for why AP fails a.k.a. ("Interpretability Illusion"), resulting in the empirical findings in prior work that sometimes a fact can be edited in an LLM while that fact is not stored at that location (layer/head) in the network in the first place. That is, the work argues that the features lie in linear subspaces that can decompose into a sum of two orthogonal vectors (`disconnected` and  `dormant`) with certain properties (Sec. 3 of the paper). This decomposition is a helpful way to think about the mechanistic underpinnings of the illusion i.e. the recent mismatch in localizing vs. editing features of LLMs.

**BEFORE rebuttal:** While this is an important topic, the major concern raised by reviewer `8jAi` and `ERkw` is writing issues and most importantly, the missing of a falsifiable hypothesis for the claim in the paper.
`ERkw` asks _"Are there a set of empirical predictions (e.g. about the directions of activations or gradients) that could either falsify or support the theoretical model?"_ and `8jAi` explicitly concerned that the paper is missing a falsifiable hypothesis.
Both the above reviewers have engaged actively in the discussions and the authors have done a diligent task of revising the paper.
The authors also explicitly stated the limitations of their work including a main limitation: _"The illusion found is somewhat specific only to subspaces of hidden MLP activations"_.

**AFTER rebuttal:** While reviewer `ERkw` is satisfied with the revision and explanations, reviewer `8jAi` is still not satisfied with the provided hypothesis, requesting a way to check whether a given subspace is `dormant`. I think an empirical answer to this question is provided in [a response](https://openreview.net/forum?id=Ebt7JgMHv1&noteId=3aC5IxMcer) to  reviewer `ERkw`:

```
To check how dormant the rowspace components are, we ran the experiment shown in Figure 12 in the paper, where we project the difference of activations of the two examples we are patching between on the causally disconnected component (the orthogonal projection on), and observe cosine similarity between this projection and the original direction of ~0.9; this implies that the cosine similarity with the rowspace component is ~0.4; thus, the rowspace component is relatively dormant compared to the nullspace one.
```

Overall, the AC agrees with the authors that they have addressed all the major concerns of the reviewers.
Most importantly, the authors have explained in the rebuttal explicitly the limitations and scope of the findings. The AC also agrees with the reviewer `8jAi` that the way for testing the hypothesis is empirical and can be significantly clarified in the paper.

**TODO:** Given the remaining concern by reviewer `8jAi`, I'd request the authors to revise the paper to make it clearer how to test the falsifiable hypothesis (and a paragraph titled **how to check for dormant subspaces**) and provide, in the Appendix, two **real** examples for when the hypothesis turns out to be (a) true and (b) false.

AC recommends `accept`.

**Justification For Why Not Higher Score:**

The paper shows an explanation for an interesting illusion in activation patching. The finding is mostly limited to the hidden MLP activations and therefore does not necessarily apply to most networks. The test for domain subspaces is empirical.

**Justification For Why Not Lower Score:**

The authors have extensively discussed the limitations of the work and concretize their contributions after the rebuttal process. The AC believes the contribution is worth publishing at ICLR.

---

### Decision · Program_Chairs · 2024-01-16

Accept (poster)